# You May Be Running the Wrong Inception Crop

## Abstract

A decade after its inception, Inception crop has become the standard crop-based data augmentation method for training deep vision models. Not only is its practice of uniformly sampling crop scale and aspect ratio widely adopted, but also its lower and upper bounds, with the scale lower bound being the sole exception that is sometimes tuned. It is therefore surprising that the standard implementation in the TensorFlow / JAX ecosystem samples crop scale with probability density function $f(A) \propto \frac{1}{\sqrt{A}}$ unlike the PyTorch counterpart, which follows the original description. Motivated by this discovery, we train 522 ViT-S/16 models on the ImageNet-1k dataset with various training budgets and crop scale distributions. We reach $78.78 \pm 0.09$ top-1 val. accuracy with 90 epochs of training budget and find that 1. Higher training budget requires stronger augmentation; 2. Lower tail of the distribution of the crop scale determines the augmentation strength of Inception crop; 3. Models trained with higher training budget exhibit sparser saliency, regardless of the crop scale distribution or weight decay. Based on 2. we revisit the performance of Beta crop, whose softer cutoff allows it to optimize model performance across training budgets with less compromise. We replicate 1. and 3. with Scion optimizer in addition to AdamW, suggesting that the results may be general.

## 1 Introduction

Originally proposed as a data augmentation method for training the Inception network (Szegedy et al., 2015) on the ImageNet-1k dataset (Deng et al., 2009), Inception crop has become the de facto standard crop-based augmentation method for training deep vision models based on CNN (convolutional neural network) (Kirichenko et al., 2023; Wightman et al., 2021; Qin et al., 2025) or ViT (vision transformer) (Beyer et al., 2022a; Steiner et al., 2022), trained on smaller (Jordan, 2024) or larger (Steiner et al., 2022) datasets, for both supervised learning and self-supervised learning (Chen et al., 2020a;b) models. Its practice of sampling crop scale[1] and aspect ratio[2] uniformly is usually taken for granted, and so are the range of the aspect ratio and the upper bound of the crop scale. The lower bound of the crop scale $a_{\min}$ is perhaps the only parameter considered tunable (Kirichenko et al., 2023), with smaller value resulting in stronger augmentation. It is therefore surprising when we track down the training loss discrepancy in reproducing the results from Beyer et al. (2022a) to the TensorFlow implementation of Inception crop with an otherwise identical setup (appendix A). Given its popularity in training deep vision models in the TensorFlow / JAX ecosystem (Chen et al., 2020a;b; Zhai et al., 2022; Steiner et al., 2022; Beyer et al., 2022a; 2023; Dehghani et al., 2024; Alabdulmohsin et al., 2024; Dahl et al., 2023), we find it critical to assess its potential impact.

Prior work has experimented on how the strength of data augmentation may impact model performance with the crop scale lower bound of the Inception crop as the main focus (Kirichenko et al., 2023) but experiments on the distribution of crop scale are more limited (Bouchacourt et al., 2021). Furthermore, both are done in isolation from each other and neither explores how varying training budget (in terms of the number of training epochs) may affect the results. It is therefore imperative to assess such implementation differences in the full context of training budget and crop scale distribution. After extensive experiments of sensible combinations of these factors, we find that:

---

[1] Crop area normalized s.t. area of the original image is 1
[2] Width of the crop divided by its height

1. Optimal augmentation strength depends on the training budget. Higher training budget requires stronger data augmentation.

2. Lower tail of the distribution of the crop scale determines the augmentation strength of Inception crop.

3. Models trained with higher training budgets exhibit sparser saliency which is unaffected by crop scale distribution or weight decay, regardless of how we measure it.

Based on 2. we revisit the performance of Beta crop, whose softer cutoff allows it to optimize model performance across training budgets with less compromise therefore partially mitigates 1.

## 2 Related Works

### 2.1 ImageNet datasets and evaluation

When introduced in 2012 for the ImageNet Large Scale Visual Recognition Challenge (ILSVRC) (Russakovsky et al., 2015), the ImageNet-1k dataset with its 1000 object classes and 1,461,406 images represented orders of magnitude jump in terms of the scale of labeled datasets. Partially due to the inaccessibility of the ImageNet-21k superset (Ridnik et al., 2021), the ImageNet-1k dataset remains the standard dataset for training and evaluating deep vision models. Nonetheless, ImageNet-1k is not without its flaws (Kisel et al., 2024) such as faulty labels (Vasudevan et al., 2022) and multi-class images (Yun et al., 2021). To mitigate the latter, Beyer et al. (2020) employ a hybrid approach using an ensemble of models and human annotators to create multi-label Reassessed Labels ("ReaL") and use them to re-evaluate progress of the field.

### 2.2 ViT training

Beyer et al. (2022a) significantly improve upon the ViT-S/16 ImageNet-1k baseline of the original ViT paper (Dosovitskiy et al., 2021) and we use their result as the baseline in our study. Beyond the scale of ImageNet-1k, Steiner et al. (2022) conduct extensive experiments and find that "AugReg" (data augmentation and model regularization) tends to benefit ViT training on smaller datasets. The reported effect of AugReg on ImageNet-21k training for 30ep vs. 300ep further hints at the general results, but Steiner et al. (2022) do not report any experiments with the parameters of Inception crop.

### 2.3 Data augmentation

Data augmentation has a long history in the field of computer vision (Simard et al., 1991) and has been considered a form of implicit regularization of the model (Hernandez-Garcia & König, 2020). In addition to crop-based augmentation, transformation-based augmentation such as RandAugment (Cubuk et al., 2020) and TrivialAugment (Muller & Hutter, 2021) and mixing-based augmentation such as Mixup (Zhang et al., 2018), CutMix (Yun et al., 2019), and AugMix (Hendrycks* et al., 2020) have all been used alone or in combination to train deep vision models. Strong data augmentation, however, can lead to confusion between co-occurring or semantically similar classes due to degradation of the semantic meaning of the image (Kirichenko et al., 2023).

### 2.4 Saliency map of deep vision models

Saliency map refers to a heatmap that highlights the most relevant regions of the image that aims to make the model's inference result more explainable. For CNNs with ReLU activation function, the local connectivity pattern and ReLU's simplicity allow simple approaches such as guided backpropagation (Springenberg et al., 2015), CAM (Zhou et al., 2016), and Grad-CAM (Selvaraju et al., 2017) to generate intuitive results. Transformer models' all-pair attention mechanism and activation functions with both positive and negative regions turn out to be more challenging for these techniques, though methods such as LRP (Binder et al., 2016) and propagated relevance map (Chefer et al., 2021) have shown some promise.

---

**Algorithm 1** `sample_distorted_bounding_box()`

---
1: **Input:** $h_o, w_o, r_{\min}, r_{\max}, a_{\min}, a_{\max}$
2: **for** $t = 1$ **to** max_attempts **do**
3:     Sample $r \sim \mathcal{U}(r_{\min}, r_{\max})$
4:     $h_{\max} = \min(\sqrt{\frac{a_{\max} h_o w_o}{r}}, h_o)$
5:     $h_{\min} = \min(\sqrt{\frac{a_{\min} h_o w_o}{r}}, h_{\max})$
6:     Sample $h \sim \mathcal{U}(h_{\min}, h_{\max})$
7:     $w = hr$
8:     $a = hw$
9:     **if** $0 < h \leq h_o$ and $0 < w \leq w_o$ and $a_{\min} h_o w_o < a \leq a_{\max} h_o w_o$ **then**
10:         Sample $y \sim \mathcal{U}(0, h_o - h)$
11:         Sample $x \sim \mathcal{U}(0, w_o - w)$
12:         Return $(y, y + h, x, x + w)$
13:     **end if**
14: **end for**
15: Return $(0, h_o, 0, w_o)$ ▷ Fallback to full image

---

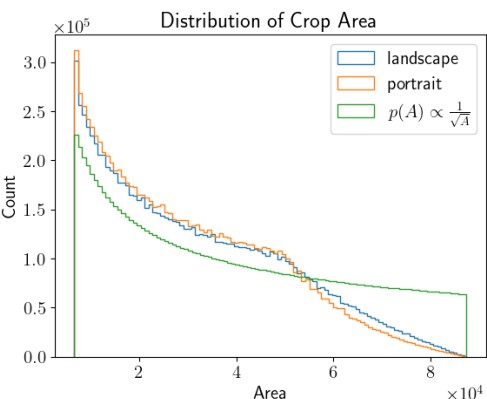

Figure 1: Crop area distribution of the standard implementation of Inception crop in TensorFlow with default parameters for $256 \times 512$ and $512 \times 256$ images, $N = 10^7$. The probability density function of the distribution is approximately proportional to $\frac{1}{\sqrt{A}}$ but truncated at larger area when some aspect ratios become inadmissible. Note that landscape and portrait images have slightly different distributions.

## 3 Inception Crop Implementations

### 3.1 TensorFlow

Standard implementation of Inception crop in TensorFlow that JAX vision libraries often depend upon (Beyer et al., 2022b; Dehghani et al., 2022) first calls `sample_distorted_bounding_box()` in `tf.image` to get the $(y_{\min}, y_{\max}, x_{\min}, x_{\max})$ bounding box for the crop. Omitting error handling and floating point arithmetic, calling `sample_distorted_bounding_box()` with image height $h_o$, image width $w_o$, aspect ratio range $[r_{\min}, r_{\max}]$, and crop scale range $[a_{\min}, a_{\max}]$ follows Algorithm 1. While seemingly reasonable, sampling the crop height $h$ from a uniform distribution means that the cumulative distribution function (CDF) of the crop scale $a$ follows $\Pr(a < A) \propto \sqrt{A} - C$, so the probability density function (PDF) follows $f(A) \propto \frac{1}{\sqrt{A}}$. We can experimentally verify this distribution (fig. 1) and examine the probability heatmaps of the crop size for both landscape and portrait images (fig. 2). Note that the probability distribution is not equivariant to image transposition since the aspect ratio is sampled uniformly in linear space and sampled $h$ is clipped as $\min(\cdot, h_o)$ instead of rejection sampling.

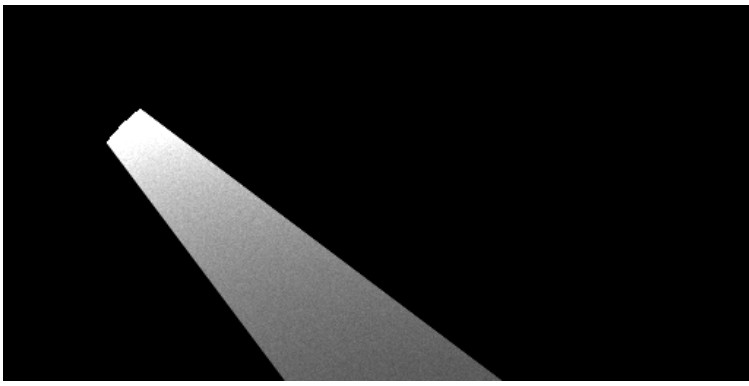

Landscape image ($h_o = 256, w_o = 512$)

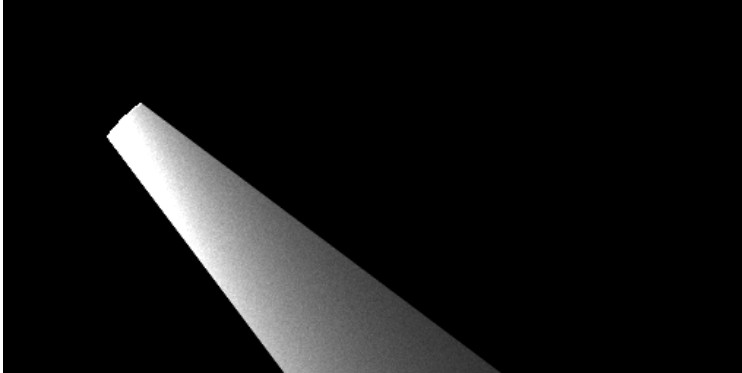

Portrait image ($h_o = 512, w_o = 256$), transposed

Figure 2: Probability heatmaps of the crop size $(h, w)$ given by the TensorFlow implementation for landscape and portrait images with default parameters $r_{\min} = \frac{3}{4}, r_{\max} = \frac{4}{3}, a_{\min} = 0.05, a_{\max} = 1, N = 10^7$.

## 3.2 PyTorch

Inception crop is included in PyTorch (Ansel et al., 2024) as part of the TorchVision library (maintainers & contributors, 2016), `RandomResizedCrop` (RRC, Kirichenko et al. (2023)) in `torchvision.transforms.v2`. Its counterpart to `sample_distorted_bounding_box()` is the static method `get_params()` which takes image tensor img, aspect ratio range $[r_{\min}, r_{\max}]$, and crop scale range $[a_{\min}, a_{\max}]$ as arguments and returns the bounding box as (top, left, $h$, $w$) following Algorithm 2. Unlike Algorithm 1, the resulting crop size distribution is indeed uniform and equivariant to image transposition (appendix B).

## 4 Beta Crop

The TensorFlow implementation of Inception crop was first committed 9 years ago and has remained essentially unchanged. The fact that the resulting crop size discrepancy goes unnoticed motivates us to test its effect on model performance and the effect of crop scale distribution on model performance in general.

Although we can generalize its practice of sampling the crop scale as $a = x^p, x \sim \mathcal{U}(a_{\min}^{\frac{1}{p}}, a_{\max}^{\frac{1}{p}})$ for $p$ other than 2, this introduces a dependency of the distribution on the lower bound of the crop scale, which may not be desirable. We therefore decide to parameterize the the crop scale distribution with the beta distribution:

$$a = a_{\min} + x(a_{\max} - a_{\min})$$
$$x \sim Beta(\alpha, \beta)$$

---

**Algorithm 2** `RandomResizedCrop.get_params()`

---

1: **Input:** img, $r_{\min}, r_{\max}, a_{\min}, a_{\max}$
2: $\_, h_o, w_o = $ `get_dimensions(img)`
3: **for** $t = 1$ **to** 10 **do**
4:      Sample $a \sim \mathcal{U}(a_{\min} h_o w_o, a_{\max} h_o w_o)$
5:      Sample $r_{\log} \sim \mathcal{U}(\log(r_{\min}), \log(r_{\max}))$
6:      $r = e^{r_{\log}}$
7:      $w = \sqrt{ar}$
8:      $h = \sqrt{\frac{a}{r}}$
9:      **if** $0 < h \leq h_o$ and $0 < w \leq w_o$ **then**
10:          Sample $y \sim \mathcal{U}(0, h_o - h)$
11:          Sample $x \sim \mathcal{U}(0, w_o - w)$
12:          Return $(y, x, h, w)$
13:      **end if**
14: **end for**
15: $r_o = \frac{w_o}{h_o}$
16: **if** $r_o < r_{\min}$ **then**
17:      $w = w_o$
18:      $h = \frac{w}{r_{\min}}$
19: **else if** $r_o > r_{\max}$ **then**
20:      $h = h_o$
21:      $w = h r_{\max}$
22: **else**
23:      $w = w_o$
24:      $h = h_o$
25: **end if**
26: Return $\left(\frac{h_o - h}{2}, \frac{w_o - w}{2}, h, w\right)$          ▷ Fallback to central crop

---

where the beta distribution $Beta(\alpha, \beta)$ is defined by the PDF

$$f(x; \alpha, \beta) = \frac{x^{\alpha-1}(1-x)^{\beta-1}}{\mathcal{B}(\alpha, \beta)}$$

with $x \in [0, 1]$ and the beta function $\mathcal{B}(\alpha, \beta)$ as the normalization constant. In addition, we clip $a_{\max}$ to be the maximum possible crop scale when the image aspect ratio $r_o$ is out of the range $[r_{\min}, r_{\max}]$ (appendix C). When $\alpha = \beta = 1$, Beta crop reduces to Inception crop as a special case (fig. 3). Although we later learn of prior experiments by Bouchacourt et al. (2021) with Beta crop, we still find it worth exploring for 3 reasons:

1. Bouchacourt et al. (2021) only train ResNet-18 on ImageNet-1k with Beta crop, which is of a different class of model with about half as many parameters (12M) as ViT-S (22M).

2. Bouchacourt et al. (2021) train ResNet-18 with fixed training budget of 150 epochs but we have seen preliminary evidence that the optimal augmentation strength may depend on the training budget.

3. Finally, Bouchacourt et al. (2021) fix $\alpha = 1$ while varying $\beta \in \{0.1, 0.5, 1, 2, 3, 10\}$, which do not cover more modest deviation from uniform distribution such as the TensorFlow Inception crop's $f(A) \propto \frac{1}{\sqrt{A}}$ or $Beta(2, 1)$.

## 5 Experiments

We train a variant of ViT-S/16 described in Beyer et al. (2022a) (sometimes called "Simple ViT", Wang) on the ImageNet-1k dataset with PyTorch as the baseline, including the full setup of 1024 batch size,

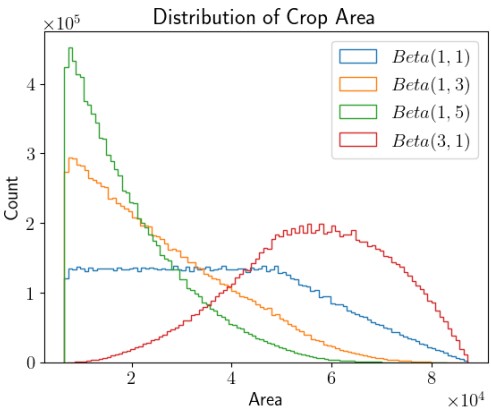

Figure 3: Crop area distribution of Beta crop with selected values of $\alpha$ and $\beta$ for landscape image ($h_o = 256, w_o = 512$), $r_{\min} = \frac{3}{4}, r_{\max} = \frac{4}{3}, a_{\min} = 0.05, a_{\max} = 1, N = 10^7$ samples for each distribution. $\alpha = \beta = 1$ results in uniform distribution as expected except truncation at higher crop scale due to inadmissible aspect ratios.

10000 warm-up steps followed by cosine learning rate (LR) decay with AdamW (Loshchilov & Hutter, 2019) optimizer, global average-pooling (GAP) (Raghu et al., 2021), 2D sin-cos positional encoding (Chen et al., 2021) and Mixup (Zhang et al., 2018) with $\alpha = 0.2$. We reimplement the RandAugment (Cubuk et al., 2020) in the Big Vision codebase (Beyer et al., 2022b) except the broken `Contrast` transform, which we elect to fix (appendix D). For convenience and consistency, we also reimplement the TensorFlow Inception crop almost line-by-line in PyTorch and verify that the PyTorch reimplementation results in indistinguishable crop size distributions from that of the original (appendix E). For both PyTorch Inception crop and reimplemented TensorFlow Inception crop, we train the models for $\{30, 60, 90, 150, 300\}$ epochs with crop scale lower bound $a_{\min} \in \{0.025, 0.05, 0.1, 0.15, 0.2, 0.25\}$. We then sweep the the parameter space of Beta crop in three disjoint sets:

1. Positive skew: $a_{\min} \in \{0.25, 0.33, 0.5\}, \alpha = 1, \beta \in \{3, 5\}$ that compensates large lower bound of crop scale with more samples near the lower bound;

2. Negative skew: $a_{\min} \in \{0.05, 0.1\}, \alpha \in \{2, 3, 5\}, \beta = 1$ that compensates small lower bound of crop scale with less samples near the lower bound.

3. Matching strength: $a_{\min} = 0, \alpha \in \{1.149, 1.287, 1.551, 1.818, 2.099, 2.403\}, \beta = 1$ that are designed to match the augmentation strength of PyTorch Inception crop with $a_{\min} \in \{0.025, 0.05, 0.1, 0.15, 0.2, 0.25\}$.

Finally, we replicate some of the results with constrained Scion optimizer (Pethick et al., 2025). We adopt the model modifications, batch size 4096, cosine LR decay with no warm-up, and maximum LR for DeiT-base (Touvron et al., 2021) from Pethick et al. (2025) and verify that the maximum LR remains optimal for the 90ep experiment by doubling and halving it. We then perform our own weight decay (WD) sweep through 4 values for all training budgets (appendix I). With the resulting setup we compare the model performance with PyTorch Inception crop and fixed $a_{\min} = 0.05$ vs. the optimized $a_{\min}$ for each training budget, identified in the AdamW experiments. For each setting we train $N = 3$ models with different random seeds. We evaluate the models on the validation set with both the original labels and the multi-label ReaL (Beyer et al., 2020). The models are trained on either 8×A100 or 2×H100 instances depending on the availability, but with standardized Python and package versions (Python 3.12.x, torch==2.5.1+cu124, and torchvision==0.20.1+cu124).

# 6 Results

## 6.1 Inception crop implementations

Here are the results of training ViT-S/16 models on ImageNet-1k with implementations of Inception crop, varying training budget and $a_{\min}$ (table 1, fig. 4, and appendix F). The optimal crop scale lower bound $a_{\min}$ shifts from $\geq 0.25$ to $\leq 0.05$ as we increase the training budget from 30 epochs to 300 epochs. Compared to the PyTorch counterpart, the (reimplemented) TensorFlow Inception crop is almost never favorable and becomes worse with smaller $a_{\min}$, likely due to the increasing positive skew of the crop scale distribution. Evaluation using the multi-label ReaL shifts the top-1 val. accuracy by $+6$–$7\%$ but the results stand (table 2, fig. 5, and appendix F). The only settings in which the TensorFlow Inception corp performs better are with 150–300ep training budget and $a_{\min} \geq 0.05$, which can be explained by the differences in augmentation strength (appendix F.3).

| $a_{\min}$ | TensorFlow Inception crop | | | | | | PyTorch Inception crop | | | | | |
|---|---|---|---|---|---|---|---|---|---|---|---|---|
| | 0.025 | 0.05 | 0.1 | 0.15 | 0.2 | 0.25 | 0.025 | 0.05 | 0.1 | 0.15 | 0.2 | 0.25 |
| 30ep | 64.59 | 65.55 | 67.03 | 68.06 | 68.78 | **69.15** | 67.08 | 67.35 | 68.26 | 68.74 | 69.26 | **69.67** |
| 60ep | 72.77 | 73.43 | 74.29 | 74.77 | 74.99 | **75.25** | 74.35 | 74.77 | 74.86 | 75.08 | 75.20 | **75.55** |
| 90ep | 75.76 | 76.25 | 76.77 | 77.03 | **77.21** | 77.16 | 76.81 | 76.92 | 77.21 | 77.21 | **77.35** | 77.16 |
| 150ep | 77.90 | 78.29 | **78.60** | 78.59 | 78.52 | 78.39 | 78.58 | 78.64 | **78.75** | 78.70 | 78.51 | 78.43 |
| 300ep | 79.43 | **79.75** | 79.64 | 79.49 | 79.20 | 79.01 | **79.80** | 79.73 | 79.62 | 79.18 | 79.13 | 78.86 |

Table 1: Mean top-1 val. accuracy (original label), TensorFlow vs. PyTorch Inception crop. As the training budget increases from 30 epochs to 300 epochs, the optimal crop scale lower bound $a_{\min}$ decreases (stronger augmentation).

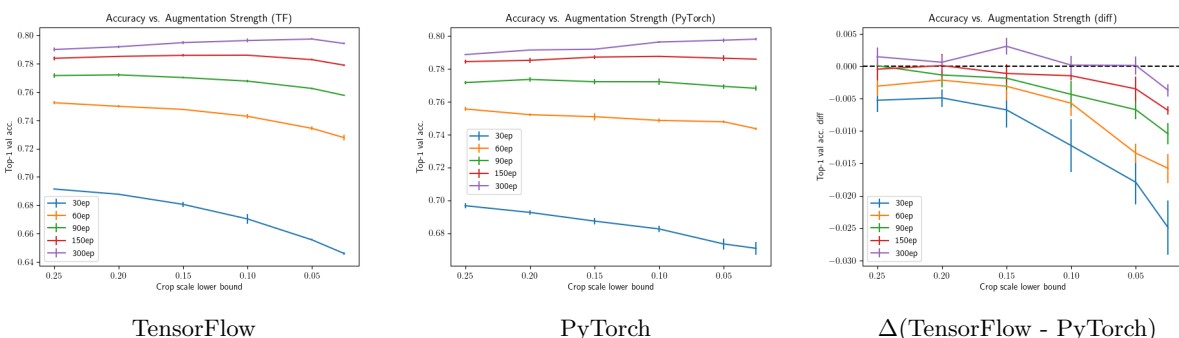

| TensorFlow | PyTorch | $\Delta$(TensorFlow - PyTorch) |
|---|---|---|

Figure 4: Top-1 val. accuracy (original label), TensorFlow vs. PyTorch Inception crop, with sample standard deviation $\sigma_{\mathrm{tf}}, \sigma_{\mathrm{torch}}, (\sigma_{\mathrm{tf}}^2 + \sigma_{\mathrm{torch}}^2)^{\frac{1}{2}}$ plotted as error bar. As the training budget increases from 30 epochs to 300 epochs, the optimal crop scale lower bound $a_{\min}$ decreases (stronger augmentation).

| $a_{\min}$ | TensorFlow Inception crop | | | | | | PyTorch Inception crop | | | | | |
|---|---|---|---|---|---|---|---|---|---|---|---|---|
| | 0.025 | 0.05 | 0.1 | 0.15 | 0.2 | 0.25 | 0.025 | 0.05 | 0.1 | 0.15 | 0.2 | 0.25 |
| 30ep | 72.23 | 73.22 | 74.77 | 75.73 | 76.45 | **76.84** | 74.77 | 75.11 | 75.90 | 76.39 | 76.88 | **77.21** |
| 60ep | 80.18 | 80.75 | 81.30 | 81.68 | 81.88 | **82.13** | 81.44 | 81.73 | 81.86 | 81.95 | 82.09 | **82.37** |
| 90ep | 82.65 | 83.10 | 83.31 | 83.57 | **83.64** | 83.56 | 83.39 | 83.39 | **83.65** | 83.58 | 83.59 | 83.50 |
| 150ep | 84.21 | 84.51 | **84.67** | 84.51 | 84.42 | 84.36 | 84.57 | **84.65** | 84.56 | 84.47 | 84.33 | 84.28 |
| 300ep | 85.30 | **85.39** | 85.25 | 85.01 | 84.71 | 84.45 | **85.32** | 85.27 | 85.06 | 84.81 | 84.59 | 84.44 |

Table 2: Mean top-1 val. accuracy (ReaL), TensorFlow vs. PyTorch Inception crop. As the training budget increases from 30 epochs to 300 epochs, the optimal crop scale lower bound $a_{\min}$ decreases (stronger augmentation).

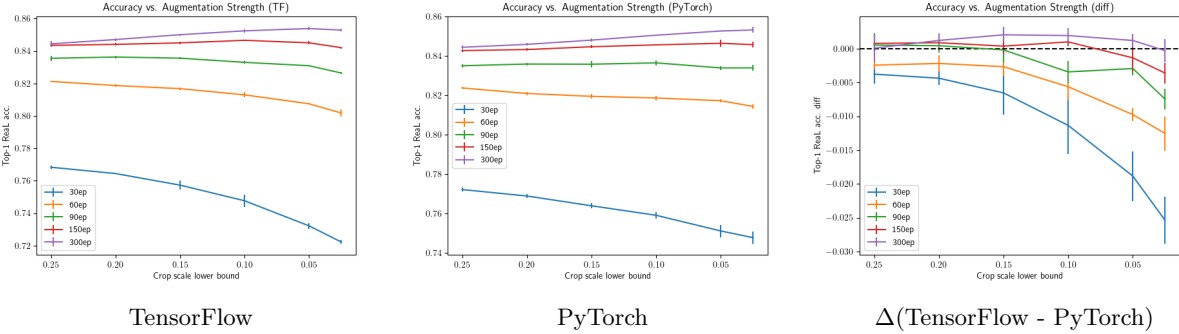

| | TensorFlow | PyTorch | $\Delta$(TensorFlow - PyTorch) |

Figure 5: Top-1 val. accuracy (ReaL), TensorFlow vs. PyTorch Inception crop, with sample standard deviation $\sigma_{\text{tf}}, \sigma_{\text{torch}}, (\sigma_{\text{tf}}^2 + \sigma_{\text{torch}}^2)^{\frac{1}{2}}$ plotted as error bar. As the training budget increases from 30 epochs to 300 epochs, the optimal crop scale lower bound $a_{\min}$ decreases (stronger augmentation).

| | $a_{\min}$ | 0.25 | 0.33 | 0.5 |
|---|---|---|---|---|
| | 30ep | 68.38±0.08 | 69.32±0.20 | **70.35±0.24** |
| | 60ep | 74.90±0.10 | 75.17±0.07 | **75.42±0.02** |
| $\beta = 3$ | 90ep | 76.69±0.11 | **76.93±0.26** | 76.70±0.12 |
| | 150ep | 78.14±0.19 | **78.18±0.14** | 77.57±0.15 |
| | 300ep | **78.85±0.14** | 78.48±0.07 | 77.63±0.03 |
| | 30ep | 67.31±0.20 | 68.92±0.24 | **70.24±0.21** |
| | 60ep | 74.01±0.05 | 74.81±0.18 | **75.42±0.20** |
| $\beta = 5$ | 90ep | 76.31±0.10 | 76.69±0.05 | **76.71±0.18** |
| | 150ep | 77.86±0.04 | **77.90±0.03** | 77.48±0.07 |
| | 300ep | **78.61±0.01** | 78.42±0.15 | 77.59±0.14 |

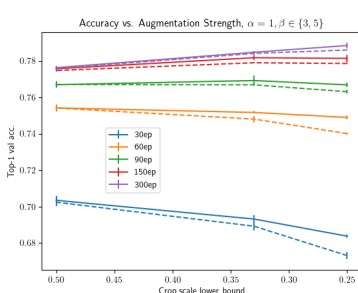

Figure 6: Top-1 val. accuracy (original label), Beta crop with $\alpha = 1$. Solid line: $\beta = 3$, Dashed line: $\beta = 5$

## 6.2 Beta crop

Among the Beta crop experiments, we find that higher positive skew of the distribution does not compensate for higher $a_{\min}$ (fig. 6). That is, none of the set of experiments with $a_{\min} = 0.25, \alpha = 1, \beta \in \{3, 5\}$ improves upon the corresponding experiment with the PyTorch Inception crop with the same $a_{\min}$. This suggests that even more samples with smaller crops in the distribution do not help the model generalize: The model needs to see out-of-distribution crops with lower crop scale.

On the other hand, negative skew of the distribution does compensate for lower $a_{\min}$ (fig. 7). That is, the set of experiments with $a_{\min} \in \{0.05, 0.1\}, \alpha \in \{2, 3, 5\}, \beta = 1$ does improve upon the corresponding experiment with the PyTorch Inception crop with the same $a_{\min}$ with training budgets 30–90 epochs. Quantitatively just like PyTorch Inception crop with $a_{\min} = 0.05$ which crops 5.3% of the samples with crop scale between 0.05 and 0.1, Beta crop with $a_{\min} = 0.05, \alpha = 2, \beta = 1$ crops 5.3% of the samples below crop scale 0.268. With the working hypothesis that the smallest 5.3% of the crops determine the model performance, we expect it to be comparable to PyTorch Inception crop with $a_{\min} = 0.227$ that also crops 5.3% of the samples below crop scale 0.268. Reexamining fig. 7 with this insight, we can see that the former indeed has performance between that of the PyTorch Inception crop with $a_{\min} = 0.2$ and 0.25 for all training budgets 30–300ep (table 1). This observation motivates the matching strength experiments below.

Beta crop with $a_{\min} = 0, \alpha \in \{1.149, 1.287, 1.551, 1.818, 2.099, 2.403\}, \beta = 1$ have the same 5th percentile crop scale as that of PyTorch Inception crop with $a_{\min} \in \{0.025, 0.05, 0.1, 0.15, 0.2, 0.25\}$, respectively. *E.g.*, $\Pr(a < 0.07375) = 0.05$ with $a \sim Beta(\alpha = 1.149, \beta = 1)$ since $a_{\min} + 0.05(1 - a_{\min}) = 0.07375$ when $a_{\min} = 0.025$. Models trained in these experiment match or exceed the performance of the PyTorch Inception crop in most settings. In particular, softer cutoff of Beta crop allows it to compromise less of the performance even when the augmentation strength is not optimally tuned for the given training budget (table 3 and fig. 8).

| | $a_{\min}$ | 0.05 | 0.1 |
|---|---|---|---|
| | 30ep | 69.68±0.04 | **70.04**±0.27 |
| | 60ep | 75.46±0.22 | **75.66**±0.15 |
| $\alpha = 2$ | 90ep | **77.35**±0.24 | 77.29±0.14 |
| | 150ep | **78.56**±0.11 | 78.28±0.17 |
| | 300ep | **79.07**±0.24 | 78.85±0.01 |
| | 30ep | 70.24±0.14 | **70.28**±0.17 |
| | 60ep | **75.58**±0.21 | 75.55±0.08 |
| $\alpha = 3$ | 90ep | 77.11±0.07 | **77.12**±0.16 |
| | 150ep | **78.15**±0.10 | 78.12±0.21 |
| | 300ep | **78.44**±0.12 | 78.23±0.12 |
| | 30ep | 70.50±0.12 | **70.66**±0.20 |
| | 60ep | **75.34**±0.29 | 75.31±0.08 |
| $\alpha = 5$ | 90ep | **76.80**±0.11 | 76.66±0.15 |
| | 150ep | **77.43**±0.11 | 77.27±0.16 |
| | 300ep | **77.60**±0.17 | 77.44±0.13 |

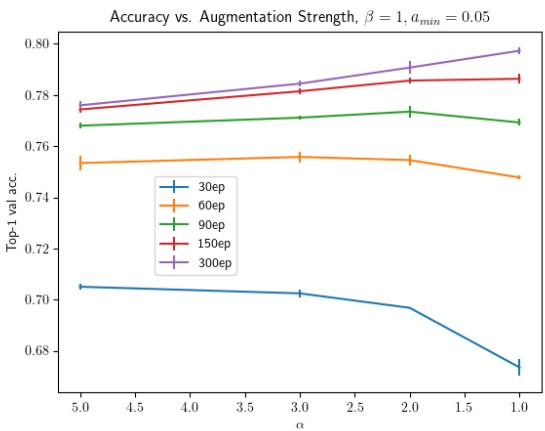

Figure 7: Top-1 val. accuracy (original label), Beta crop with $\beta = 1$. Top-1 val. accuracy vs. $\alpha$, $a_{\min} = 0.05$ is plotted on the right. For $\alpha = \beta = 1$, we reuse the data points of the PyTorch Inception crop experiments.

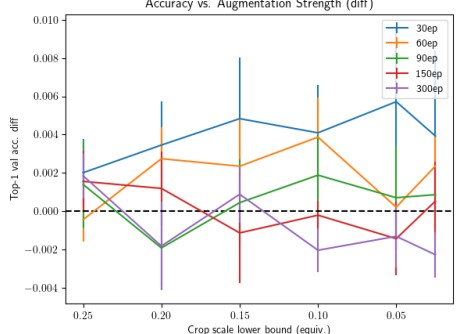
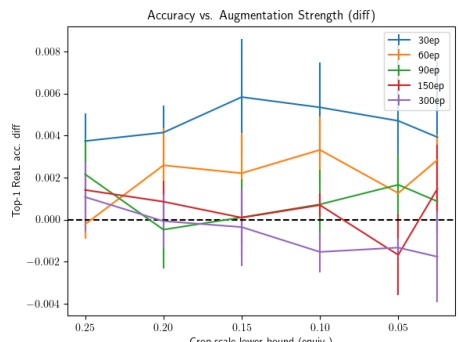

Figure 8: Diff. in top-1 val. accuracy (Beta crop, matching strength - PyTorch), original label (left) and ReaL (right), with $(\sigma_{\text{beta}}^2 + \sigma_{\text{torch}}^2)^{\frac{1}{2}}$ plotted as error bar.

| | Original label | | | | | | ReaL | | | | | |
|---|---|---|---|---|---|---|---|---|---|---|---|---|
| $\alpha$ | 1.149 | 1.287 | 1.551 | 1.818 | 2.099 | 2.403 | 1.149 | 1.287 | 1.551 | 1.818 | 2.099 | 2.403 |
| 30ep | 67.47 | 67.92 | 68.67 | 69.22 | 69.61 | **69.87** | 75.16 | 75.58 | 76.44 | 76.97 | 77.30 | **77.58** |
| 60ep | 74.59 | 74.79 | 75.24 | 75.32 | 75.48 | **75.51** | 81.72 | 81.85 | 82.20 | 82.17 | **82.35** | 82.35 |
| 90ep | 76.89 | 76.99 | **77.39** | 77.25 | 77.15 | 77.29 | 83.48 | 83.56 | **83.72** | 83.59 | 83.54 | 83.72 |
| 150ep | 78.63 | 78.49 | **78.73** | 78.59 | 78.63 | 78.59 | **84.72** | 84.48 | 84.63 | 84.48 | 84.41 | 84.42 |
| 300ep | 79.57 | **79.60** | 79.42 | 79.27 | 78.95 | 79.04 | **85.15** | 85.14 | 84.90 | 84.77 | 84.58 | 84.55 |

Table 3: Mean top-1 val. accuracy of Beta crop, matching strength, original label (left) and ReaL (right). As the training budget increases from 30 epochs to 300 epochs, the optimal $\alpha$ decreases (stronger augmentation). $\alpha = 1.149$ may be an exception possibly due to too many crops near zero area. In future iterations we may want small but nonzero $a_{\min}$ (say $0 < a_{\min} \leq 0.025$) instead while keeping the same 5th percentile crop scale.

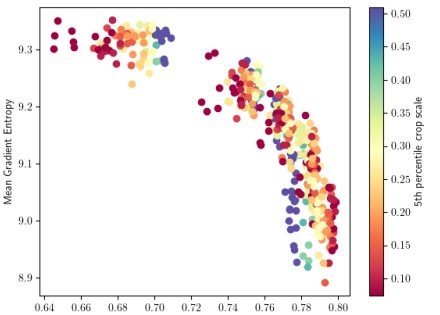 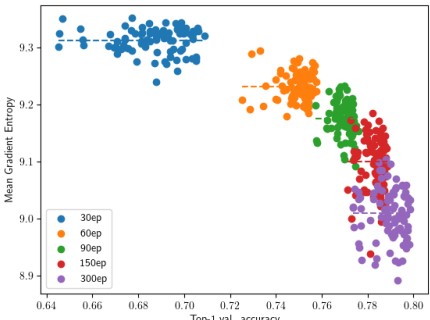

Figure 9: Mean gradient entropy $\overline{S_{\mathrm{grad}}}$ defined in eq. (2) vs. top-1 val. accuracy for all 450 Inception crop and Beta crop experiments with AdamW optimizer, color-coded by the 5th percentile crop scale (left) and training budget (right). For reference and sanity check, $S_{\mathrm{grad}} \leq \log(224^2) = 10.82$ given our input resolution. Dashed lines mark the average of the experiments of the given training budget.

For all of the Beta crop experiments the correlation between the optimal augmentation strength and training budget continues to hold, and evaluating the models using the multi-label ReaL does not change the results (appendix G).

## 6.3 Saliency sparsity vs. training budget

Since the crop scale determines the proportion of the image on which the model is trained, we initially hypothesize that crop scale distribution may affect saliency sparsity of the model. We first measure it using the rectified gradient of the class logit on the pixels as the proxy. That is, given the logit $y^c$ of the target class $c$ returned by the model, we compute its gradient on the pixels $a_{\mathrm{ijk}}$, summed over the RGB channels $k \in [0 .. 2]$ and clipped as $\max(0, \cdot)$:

$$g_{\mathrm{ij}} = \max(0, \sum_k \frac{\partial y^c}{\partial a_{\mathrm{ijk}}}) \tag{1}$$

where $i \in [0 .. h - 1]$ and $j \in [0 .. w - 1]$, $h = w = 224$ for our input resolution. We can then measure the entropy of the distribution of the rectified gradient after normalization:

$$S_{\mathrm{grad}} = -\sum_{\mathrm{ij}} g'_{\mathrm{ij}} \log g'_{\mathrm{ij}}, \quad g'_{\mathrm{ij}} = \frac{g_{\mathrm{ij}}}{\sum_{\mathrm{ij}} g_{\mathrm{ij}}} \tag{2}$$

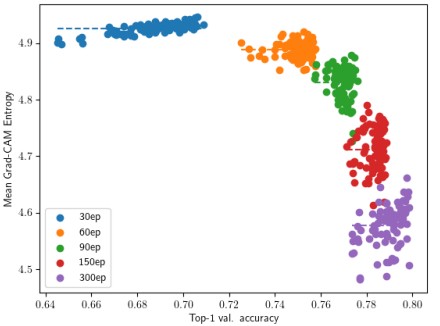 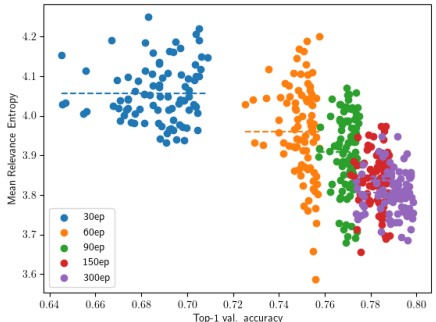

Figure 10: Mean Grad-CAM entropy $\overline{S_{\text{grad-cam}}}$ (left) and mean relevance entropy $\overline{S_{\text{rel}}}$ (right) vs. top-1 val. accuracy for all 450 Inception crop and Beta crop experiments with AdamW optimizer, color-coded by the training budget. For reference and sanity check, $S_{\text{grad-cam}}, S_{\text{rel}} \leq \log(14^2) = 5.28$ given our input resolution and patch size. Dashed lines mark the average of the experiments of the given training budget.

Using the 50000 images and target class of the ImageNet-1k validation, we find no correlation between the mean gradient entropy $\overline{S_{\text{grad}}}$ and the crop scale distribution with which the model is trained. Instead, it correlates with the training budget (fig. 9). If we inspect each individual model, we can see that the distribution of $S_{\text{grad}}$ for the 50000 validation set images shifts to the left as we increase the training budget (appendix H.1). We repeat the measurement with Grad-CAM (Selvaraju et al., 2017) and LRP of the tokens of the last layer (Binder et al. (2016), appendix H.3) and compute the entropy of their distribution, $S_{\text{grad-cam}} = -\sum_{ij} L_{ij}^c \log L_{ij}^c$ and $S_{\text{rel}} = -\sum_{ij} R_{ij}' \log R_{ij}'$, $i, j \in [0..13]$, after normalization. We find that the result stands: The entropy of saliency always decreases as we increase the training budget (fig. 10).

## 6.4 Partial replication with Scion optimizer

Finally, we replicate the spontaneous sparsification of saliency and the correlation between optimal augmentation strength and training budget with constrained Scion (Pethick et al., 2025) optimizer instead of AdamW. As expected from table 2, PyTorch Inception crop with $a_{\min} = 0.25, 0.25, 0.1, 0.025$ outperform $a_{\min} = 0.05$ for training budgets 30ep, 60ep, 90ep, and 300ep respectively (table 4). The entropy of saliency also generally decreases as we increase the training budget for the Scion experiments (fig. 11).

|  | Original label | | ReaL | |
|---|---|---|---|---|
|  | baseline | optimized | baseline | optimized |
| 30ep | 73.31±0.09 | **74.46**±0.19 | 80.83±0.07 | **81.70**±0.08 |
| 60ep | 77.44±0.09 | **77.76**±0.15 | 84.09±0.10 | **84.16**±0.16 |
| 90ep | 78.68±0.09 | **78.78**±0.09 | 84.94±0.05 | **84.96**±0.07 |
| 150ep | 79.65±0.07 | — | 85.41±0.05 | — |
| 300ep | 80.10±0.14 | **80.23**±0.17 | 85.66±0.21 | **85.71**±0.15 |

Table 4: Top-1 val. accuracy of Scion experiments, $a_{\min} = 0.05$ baseline vs. optimized $a_{\min}$ for the training budget, original label (left) and ReaL (right). As the training budget increases from 30 epochs to 300 epochs, the optimal crop scale lower bound $a_{\min}$ decreases (stronger augmentation) while $a_{\min} = 0.05$ is already expected to be optimal for 150ep.

## 7 Conclusions

In conclusion, we find that

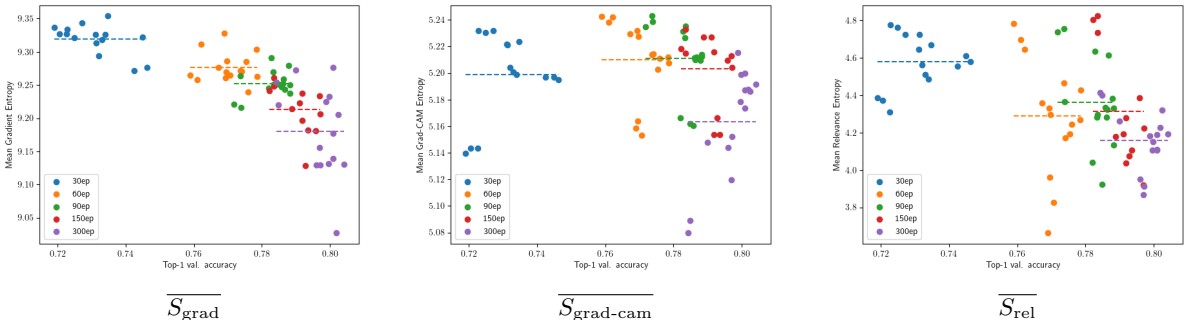

$$\overline{S_{\mathrm{grad}}} \qquad\qquad \overline{S_{\mathrm{grad\text{-}cam}}} \qquad\qquad \overline{S_{\mathrm{rel}}}$$

Figure 11: Mean gradient entropy $\overline{S_{\mathrm{grad}}}$ (left), mean Grad-CAM entropy $\overline{S_{\mathrm{grad\text{-}cam}}}$ (middle), and mean relevance entropy $\overline{S_{\mathrm{rel}}}$ (right) vs. top-1 val. accuracy for all 72 Scion experiments including that of table 4 and WD sweep (table 18), color-coded by training budget.

1. Optimal augmentation strength depends on the training budget. Higher training budget requires stronger data augmentation (section 6.1).

2. Lower tail of the distribution of the crop scale determines the augmentation strength of Inception crop (section 6.2).

3. Models trained with higher training budget exhibit sparser saliency, regardless of the augmentation strength (sections 6.3 and 6.4) or weight decay (appendix I).

In practice, 1. can further complicate efforts to make training budget more flexible (Hägele et al., 2024) or schedule-free (Defazio et al., 2024) for deep vision models. Generalized curriculum learning (Wang et al., 2024) though may be a viable method of mitigating these complications. Another research direction that may be worth pursuing is whether 1. changes the trade-off between model size and number of training epochs. *E.g.*, it may be advantageous to opt for training a larger model for fewer epochs with weaker augmentation, especially if we want to alleviate the negative effects of strong data augmentation due to semantic degradation (Kirichenko et al., 2023). Our partial mitigation in light of 2. is to propose Beta crop with matching strength whose softer cutoff allows it to optimize model performance across training budgets with less compromise (table 3 and fig. 8). The TensorFlow implementation of the Inception crop is (likely unintentionally) the opposite with positive skew that increases with lower $a_{\min}$ and consequently underperforms in most settings (tables 1 and 2) even if we adopt a unified augmentation strength measure based on 2. (appendix F.3). We therefore recommend the TensorFlow / JAX community to fix or replace their implementation and urge caution while comparing results between the TensorFlow / JAX vs. the PyTorch ecosystems for models trained with "Inception crop".

From the theoretical perspective, it is not clear why 1. and 3. are true. One possibility for 1. is that stronger augmentation helps delay memorization of limited training examples, which would otherwise prevent further generalization (Bayat et al., 2025). Assumed that conversely delaying memorization past the end of the training budget leads to underfitting, the optimal augmentation needs to be tuned such that memorization happens at the end, leading to our observation.

## 8 Limitations

The main limitations of our study are due to our focus on training ViT-S/16 models on the ImageNet-1k dataset, so it leaves the question open whether the results generalize to other model architectures, datasets, or larger/smaller models. However, it is worth pointing out again that Steiner et al. (2022) show in Figure 4 that AugReg becomes helpful for training a wide variety of ViT and ResNet models on ImageNet-21k as we increase the training budget from 30ep to 300ep, thus lending support to conclusion 1. Similarly, Bouchacourt et al. (2021) report in Section 2.2 that Beta crop with $\alpha = 1, \beta \in \{0.1, 2, 3, 10\}$ leads to degraded performance of ResNet-18 trained on ImageNet-1k for 150 epochs, in support of conclusion 2.

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

## A  TensorFlow vs. PyTorch Inception crop training loss

Figure 12 shows the training loss discrepancy when we train the linear head variant of ViT-S/16 from Beyer et al. (2022a) for 90 epochs. Since the TensorFlow Inception crop used by Big Vision oversamples smaller crops, it consistently reports higher training loss.

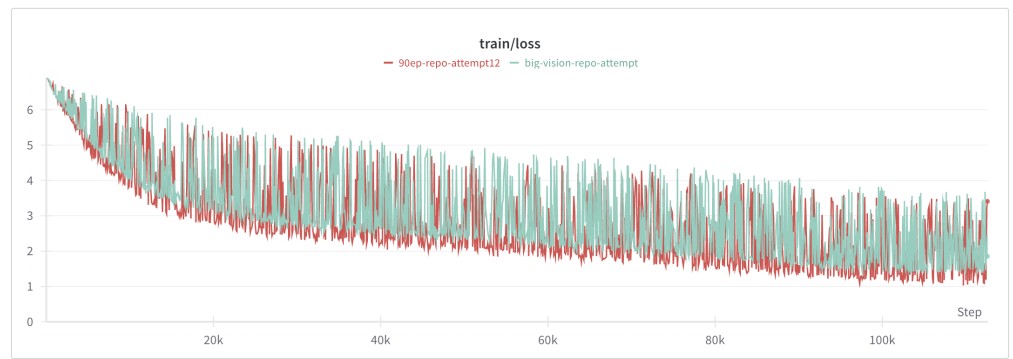

Figure 12: Training loss of the linear head variant of ViT-S/16 from Beyer et al. (2022a) for 90 epochs using the Big Vision (Beyer et al., 2022b) first-party code (cyan) vs. our reimplementation in TorchVision (maintainers & contributors, 2016) (red).

## B  Crop size distribution of the PyTorch Inception crop

Here is the probability heatmap of the crop size $(h, w)$ given by the PyTorch Inception crop for landscape and portrait images (fig. 13). About $1.4 \times 10^4$ out of $N = 10^7$ calls of RRC fail to sample a constraint-respecting crop within the hard-coded `max_attempts=10` and fall back to central crop, but otherwise the distribution is uniform and equivariant to image transposition.

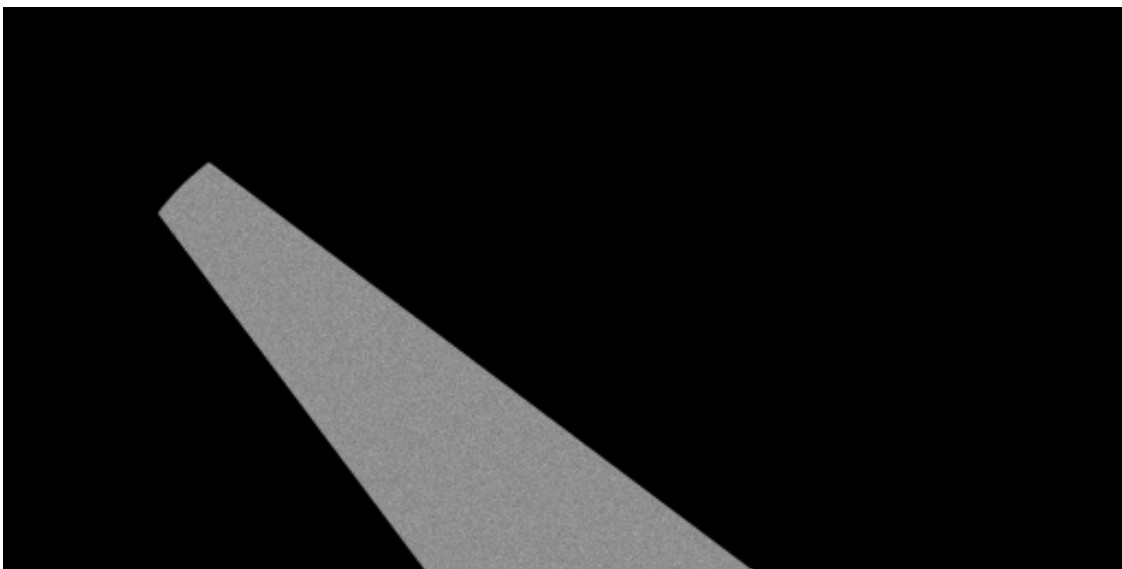

Figure 13: Probability heatmap of the crop size $(h, w)$ given by the PyTorch Inception crop (`RandomResizedCrop`, RRC) for landscape image $(h_o = 256, w_o = 512)$ with the same parameters as in fig. 2: $r_{\min} = \frac{3}{4}, r_{\max} = \frac{4}{3}, a_{\min} = 0.05, a_{\max} = 1, N = 10^7$. The counterpart for portrait image $(h_o = 512, w_o = 256$ is indistinguishable after image transposition.

## C  Beta crop implementation

The source code of our implementation of Beta crop is available here which follows algorithm 3. The part that differs from algorithm 2 is highlighted in red. Clipping $A_{\max}$ reduces sampling rejections and makes sure that we use the full support $x \in [0, 1]$ of the beta distribution.

---

**Algorithm 3** `BetaCrop`

---

1: **Input:** img, $\alpha, \beta$, $r_{\min}, r_{\max}, a_{\min}, a_{\max}$
2: $\_, h_o, w_o = $ `get_dimensions`(img)
3: $A_{\min} = a_{\min} h_o w_o$
4: $A_{\max} = a_{\max} h_o w_o$
5: **if** $r_o > r_{\max}$ **then**
6:     $A_{\max} = \min(A_{\max}, r_{\max} h_o^2)$
7: **else if** $r_o < r_{\min}$ **then**
8:     $A_{\max} = \min(A_{\max}, \frac{w_o^2}{r_{\min}})$
9: **end if**
10: **for** $t = 1$ **to** max_attempts **do**
11:     Sample $r_{\log} \sim \mathcal{U}(\log(r_{\min}), \log(r_{\max}))$
12:     $r = e^{r_{\log}}$
13:     Sample $x \sim Beta(\alpha, \beta)$
14:     $a = A_{\min} + x(A_{\max} - A_{\min})$
15:     $w = \sqrt{ar}$
16:     $h = \sqrt{\frac{a}{r}}$
17:     **if** $0 < h \le h_o$ and $0 < w \le w_o$ **then**
18:         Sample $y \sim \mathcal{U}(0, h_o - h)$
19:         Sample $x \sim \mathcal{U}(0, w_o - w)$
20:         Return $(y, x, h, w)$
21:     **end if**
22: **end for**
23: **if** $r_o < r_{\min}$ **then**
24:     $w = w_o$
25:     $h = \frac{w}{r_{\min}}$
26: **else if** $r_o > r_{\max}$ **then**
27:     $h = h_o$
28:     $w = h r_{\max}$
29: **else**
30:     $w = w_o$
31:     $h = h_o$
32: **end if**
33: Return $(\frac{h_o - h}{2}, \frac{w_o - w}{2}, h, w)$         ▷ Fallback to central crop

---

## D   RandAugment implementation

In contrast to the original RandAugment paper (Cubuk et al., 2020) which describes RandAugment as having the following 14 transforms:

- Identity
- AutoContrast
- Equalize
- Rotate
- Solarize

- Color
- Posterize
- Contrast
- Brightness
- Sharpness

- ShearX
- ShearY
- TranslateX
- TranslateY

The `efficientnet` repository linked from the paper implements RandAugment with a lineup of 16 transforms:

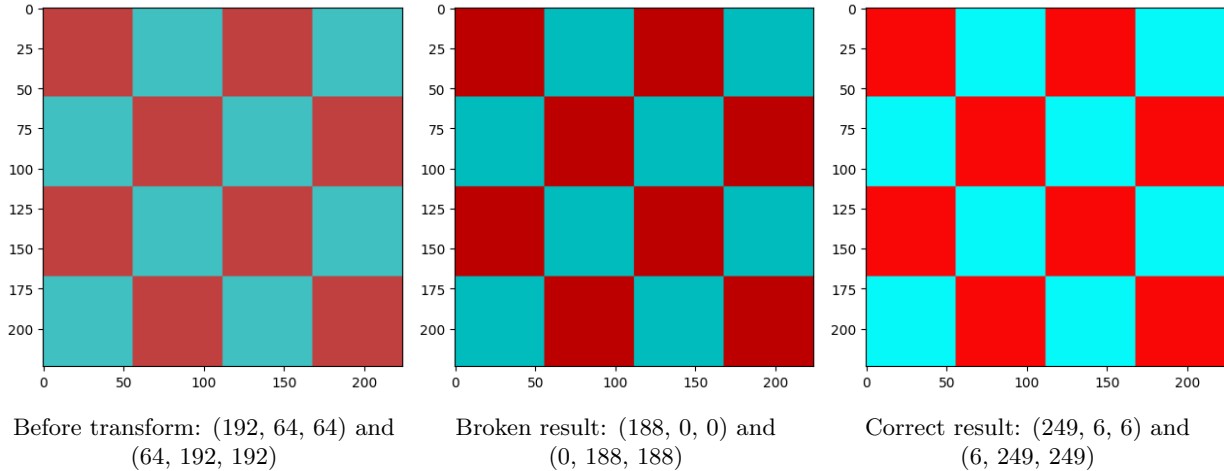

Before transform: (192, 64, 64) and (64, 192, 192)  Broken result: (188, 0, 0) and (0, 188, 188)  Correct result: (249, 6, 6) and (6, 249, 249)

Figure 14: Color grids, before and after calling `Contrast(·, 1.9)` and their RGB values.

- AutoContrast
- Equalize
- Rotate
- Solarize
- Color
- Posterize

- Contrast
- Brightness
- Sharpness
- ShearX
- ShearY
- TranslateX

- TranslateY
- Invert
- Cutout
- SolarizeAdd

In addition, the `Contrast` transform is broken. In short, when trying to compute the average RGB values, it accidentally computes $\frac{hw}{256}$ ,where $hw$ is the number of pixels, instead of the weighted average of the binned values. As a result, it tends to either brighten or darken the image instead of enhancing or reducing the contrast (fig. 14). We reimplement the 16-transform lineup following the source code of the repository but fix the `Contrast` transform in PyTorch (source code).

# E   Reimplementation of the TensorFlow Inception crop in PyTorch

For convenience and consistency, we reimplement the TensorFlow Inception crop in PyTorch (source code) and use it in our experiments instead of switching to TensorFlow or Jax. We have verified that both the crop area distribution (fig. 15) and the probability heat maps are indistinguishable from that of the original TensorFlow Inception crop.

| $a_{\min}$ | 0.025 | 0.05 | 0.1 | 0.15 | 0.2 | 0.25 |
|---|---|---|---|---|---|---|
| 30ep | 64.59±0.11 | 65.55±0.07 | 67.03±0.36 | 68.06±0.19 | 68.78±0.05 | **69.15**±0.06 |
| 60ep | 72.77±0.21 | 73.43±0.13 | 74.29±0.15 | 74.77±0.05 | 74.99±0.08 | **75.25**±0.10 |
| 90ep | 75.76±0.04 | 76.25±0.05 | 76.77±0.09 | 77.03±0.08 | **77.21**±0.13 | 77.16±0.16 |
| 150ep | 77.90±0.07 | 78.29±0.06 | **78.60**±0.06 | 78.59±0.10 | 78.52±0.06 | 78.39±0.13 |
| 300ep | 79.43±0.07 | **79.75**±0.07 | 79.64±0.14 | 79.49±0.12 | 79.20±0.10 | 79.01±0.14 |

Table 5: Top-1 val. accuracy (original label), TensorFlow Inception crop. As the training budget increases from 30 epochs to 300 epochs, the optimal crop scale lower bound $a_{\min}$ decreases (stronger augmentation).

| $a_{\min}$ | 0.025 | 0.05 | 0.1 | 0.15 | 0.2 | 0.25 |
|---|---|---|---|---|---|---|
| 30ep | 67.08±0.41 | 67.35±0.33 | 68.26±0.20 | 68.74±0.20 | 69.26±0.13 | **69.67**±0.17 |
| 60ep | 74.35±0.08 | 74.77±0.08 | 74.86±0.13 | 75.08±0.22 | 75.20±0.07 | **75.55**±0.11 |
| 90ep | 76.81±0.16 | 76.92±0.13 | 77.21±0.19 | 77.21±0.17 | **77.35**±0.14 | 77.16±0.09 |
| 150ep | 78.58±0.02 | 78.64±0.18 | **78.75**±0.04 | 78.70±0.10 | 78.51±0.17 | 78.43±0.11 |
| 300ep | **79.80**±0.06 | 79.73±0.12 | 79.62±0.05 | 79.18±0.03 | 79.13±0.04 | 78.86±0.01 |

Table 6: Top-1 val. accuracy (original label), PyTorch Inception crop. As the training budget increases from 30 epochs to 300 epochs, the optimal crop scale lower bound $a_{\min}$ decreases (stronger augmentation).

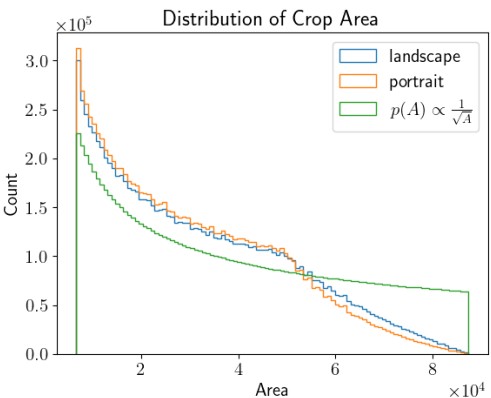

Figure 15: Crop area distribution of the reimplementation of TensorFlow Inception crop in PyTorch with default parameters for $256 \times 512$ and $512 \times 256$ images, $N = 10^7$. The results are indistinguishable from fig. 1.

## F   Detailed Inception crop results

### F.1   Full tables with error bars

Tables 5 to 8 show the full results for training ViT-S/16 models on ImageNet-1k with implementations of Inception crop, varying training budget and $a_{\min}$. We train $N = 3$ for each setting and values are given as (mean) ± (sample standard deviation), both rounded to 2 decimal places.

### F.2   Statistical significance

Tables 9 and 10 show the statistical significance of the differences in top-1 val. accuracies between the TensorFlow and PyTorch Inception crop with various training budgets while tables 11 and 12 show the statistical significance of the differences in top-1 val. accuracies across different values of crop scale lower bound $a_{min}$ for the TensorFlow Inception crop and tables 13 and 14 show the statistical significance of the differences in top-1 val. accuracies across different values of crop scale lower bound $a_{min}$ for the PyTorch Inception crop.

| $a_{\min}$ | 0.025 | 0.05 | 0.1 | 0.15 | 0.2 | 0.25 |
|---|---|---|---|---|---|---|
| 30ep | 72.23±0.13 | 73.22±0.18 | 74.77±0.39 | 75.73±0.28 | 76.45±0.02 | **76.84**±0.11 |
| 60ep | 80.18±0.23 | 80.75±0.02 | 81.30±0.15 | 81.68±0.08 | 81.88±0.08 | **82.13**±0.03 |
| 90ep | 82.65±0.05 | 83.10±0.03 | 83.31±0.08 | 83.57±0.06 | **83.64**±0.07 | 83.56±0.16 |
| 150ep | 84.21±0.06 | 84.51±0.10 | **84.67**±0.03 | 84.51±0.08 | 84.42±0.07 | 84.36±0.10 |
| 300ep | 85.30±0.09 | **85.39**±0.09 | 85.25±0.11 | 85.01±0.07 | 84.71±0.09 | 84.45±0.19 |

Table 7: Top-1 val. accuracy (ReaL), TensorFlow Inception crop. As the training budget increases from 30 epochs to 300 epochs, the optimal crop scale lower bound $a_{\min}$ decreases (stronger augmentation).

| $a_{\min}$ | 0.025 | 0.05 | 0.1 | 0.15 | 0.2 | 0.25 |
|---|---|---|---|---|---|---|
| 30ep | 74.77±0.33 | 75.11±0.32 | 75.90±0.18 | 76.39±0.14 | 76.88±0.10 | **77.21**±0.09 |
| 60ep | 81.44±0.11 | 81.73±0.09 | 81.86±0.12 | 81.95±0.11 | 82.09±0.09 | **82.37**±0.05 |
| 90ep | 83.39±0.15 | 83.39±0.09 | **83.65**±0.14 | 83.58±0.17 | 83.59±0.07 | 83.50±0.08 |
| 150ep | 84.57±0.14 | **84.65**±0.17 | 84.56±0.05 | 84.47±0.08 | 84.33±0.06 | 84.28±0.07 |
| 300ep | **85.32**±0.15 | 85.27±0.03 | 85.06±0.01 | 84.81±0.09 | 84.59±0.06 | 84.44±0.08 |

Table 8: Top-1 val. accuracy (ReaL), PyTorch Inception crop. As the training budget increases from 30 epochs to 300 epochs, the optimal crop scale lower bound $a_{\min}$ decreases (stronger augmentation).

| $a_{min}$ | 0.025 | 0.05 | 0.1 | 0.15 | 0.2 | 0.25 |
|---|---|---|---|---|---|---|
| 30ep | 5.1e-4 | 8.0e-4 | 6.6e-3 | 0.013 | 3.3e-3 | 7.5e-3 |
| 60ep | 2.6e-4 | 1.e-4 | 7.3e-3 | 0.074 | 0.029 | 0.024 |
| 90ep | 4.0e-4 | 1.2e-3 | 0.022 | 0.16 | 0.29 | 0.93 |
| 150ep | 6.6e-5 | 0.034 | 0.021 | 0.25 | 0.91 | 0.68 |
| 300ep | 2.5e-3 | 0.86 | 0.84 | 0.013 | 0.36 | 0.14 |

Table 9: T-test p-values of the differences in top-1 val. accuracies (original) between the TensorFlow and PyTorch Inception crop, two-sided, equal variance, color-coded ( $p < 0.05$ , $p < 0.01$ , $p < 0.001$ ).

| $a_{min}$ | 0.025 | 0.05 | 0.1 | 0.15 | 0.2 | 0.25 |
|---|---|---|---|---|---|---|
| 30ep | 2.3e-4 | 8.8e-4 | 0.01 | 0.024 | 1.6e-3 | 0.011 |
| 60ep | 1.0e-3 | 5.7e-5 | 7.6e-3 | 0.027 | 0.034 | 2.4e-3 |
| 90ep | 1.1e-3 | 7.2e-3 | 0.022 | 0.88 | 0.47 | 0.63 |
| 150ep | 0.015 | 0.31 | 0.04 | 0.59 | 0.18 | 0.33 |
| 300ep | 0.79 | 0.091 | 0.036 | 0.035 | 0.1 | 0.95 |

Table 10: T-test p-values of the differences in top-1 val. accuracies (ReaL) between the TensorFlow and PyTorch Inception crop, two-sided, equal variance, color-coded ( $p < 0.05$ , $p < 0.01$ , $p < 0.001$ ).

| $a_{min}$ | 0.025 | 0.05 | 0.1 | 0.15 | 0.2 | 0.25 |
|---|---|---|---|---|---|---|
| 30ep | 1.8e-4 | 1 | 2.1e-3 | 2.6e-5 | 3.e-7 | 2.5e-7 |
| 60ep | 9.8e-3 | 1 | 1.7e-3 | 7.1e-5 | 6.2e-5 | 4.4e-5 |
| 90ep | 1.9e-4 | 1 | 9.4e-4 | 1.5e-4 | 2.7e-4 | 7.0e-4 |
| 150ep | 1.8e-3 | 1 | 2.9e-3 | 9.9e-3 | 9.8e-3 | 0.31 |
| 300ep | 5.e-3 | 1 | 0.3 | 0.037 | 1.3e-3 | 1.2e-3 |

Table 11: T-test p-values of the top-1 val. accuracies (original) between the default $a_{min} = 0.05$ and other values with various training budgets, TensorFlow Inception crop, two-sided, equal variance, color-coded ( $p < 0.05$ , $p < 0.01$ , $p < 0.001$ ).

## F.3 Comparison at equal augmentation strength

Even though both are usually considered "Inception crop" and we compare them at the same crop scale lower bound $a_{\min}$, the difference in crop scale distribution between the TensorFlow and PyTorch Inception

| $a_{min}$ | 0.025 | 0.05 | 0.1 | 0.15 | 0.2 | 0.25 |
|-----------|-------|------|-----|------|-----|------|
| 30ep  | 1.5e-3 | 1 | 3.4e-3 | 2.1e-4 | 7.4e-6 | 8.5e-6 |
| 60ep  | 0.013  | 1 | 3.7e-3 | 3.4e-5 | 2.3e-5 | 5.2e-7 |
| 90ep  | 1.8e-4 | 1 | 0.016  | 2.8e-4 | 3.1e-4 | 8.5e-3 |
| 150ep | 0.013  | 1 | 0.071  | 0.96   | 0.26   | 0.13   |
| 300ep | 0.27   | 1 | 0.16   | 4.6e-3 | 7.3e-4 | 1.4e-3 |

Table 12: T-test p-values of the top-1 val. accuracies (ReaL) between the default $a_{min} = 0.05$ and other values with various training budget, TensorFlow Inception crop, two-sided, equal variance, color-coded ( $p < 0.05$ , $p < 0.01$ , $p < 0.001$ ).

| $a_{min}$ | 0.025 | 0.05 | 0.1 | 0.15 | 0.2 | 0.25 |
|-----------|-------|------|-----|------|-----|------|
| 30ep  | 0.43   | 1 | 0.016 | 3.4e-3 | 7.3e-4 | 4.3e-4 |
| 60ep  | 2.7e-3 | 1 | 0.38  | 0.081  | 2.1e-3 | 5.3e-4 |
| 90ep  | 0.39   | 1 | 0.1   | 0.083  | 0.019  | 0.066  |
| 150ep | 0.62   | 1 | 0.34  | 0.6    | 0.42   | 0.17   |
| 300ep | 0.45   | 1 | 0.21  | 1.5e-3 | 1.2e-3 | 2.2e-4 |

Table 13: T-test p-values of the top-1 val. accuracies (original) between $a_{min} = 0.05$ and other values with various training budgets, PyTorch Inception crop, two-sided, equal variance, color-coded ( $p < 0.05$ , $p < 0.01$ , $p < 0.001$ ).

| $a_{min}$ | 0.025 | 0.05 | 0.1 | 0.15 | 0.2 | 0.25 |
|-----------|-------|------|-----|------|-----|------|
| 30ep  | 0.27  | 1 | 0.019  | 3.1e-3 | 7.4e-4 | 3.8e-4 |
| 60ep  | 0.023 | 1 | 0.18   | 0.055  | 6.9e-3 | 4.3e-4 |
| 90ep  | 0.98  | 1 | 0.057  | 0.17   | 0.042  | 0.19   |
| 150ep | 0.6   | 1 | 0.47   | 0.18   | 0.039  | 0.027  |
| 300ep | 0.56  | 1 | 2.6e-4 | 1.1e-3 | 4.9e-5 | 8.6e-5 |

Table 14: T-test p-values of the top-1 val. accuracies (ReaL) between $a_{min} = 0.05$ and other values with various training budget, PyTorch Inception crop, two-sided, equal variance, color-coded ( $p < 0.05$ , $p < 0.01$ , $p < 0.001$ ).

crop leaves the question open whether the discrepancy in model performance is merely due to different augmentation strength. Figure 16 shows that TensorFlow Inception crop generally underperforms even when we use the 5th percentile crop scale as the unified measure of the augmentation strength, with reasonable interpolation.

## G  Detailed Beta crop results

### G.1  Positive / Negative skew results with ReaL

Figures 17 and 18 show the ReaL evaluation results of the ViT-S/16 models trained on ImageNet-1k with Beta crop, positive / negative skew. We train $N = 3$ for each setting and values are given as (mean) $\pm$ (sample standard deviation), both rounded to 2 decimal places.

### G.2  Matching strength, full tables with error bars

Tables 15 and 16 show the full evaluation results of the ViT-S/16 models trained on ImageNet-1k with Beta crop, matching strength, original label (table 15) and ReaL (table 16). We train $N = 3$ for each setting and values are given as (mean) $\pm$ (sample standard deviation), both rounded to 2 decimal places.

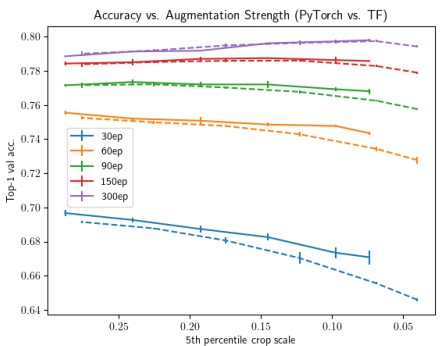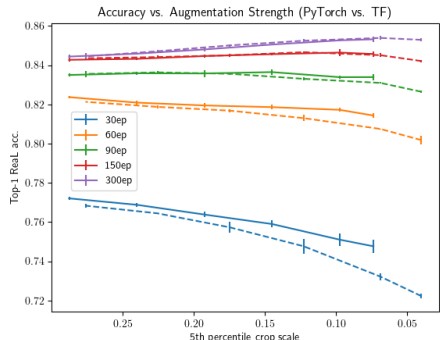

Figure 16: Top-1 val. accuracy of PyTorch Inception crop (solid line) vs. TensorFlow Inception crop (dashed line), original label (left) and ReaL (right), plotted with the 5th percentile crop scale as the x-axis. TensorFlow Inception crop generally underperforms except 300ep where it matches PyTorch Inception crop even with such unified augmentation strength measure.

| | $a_{min}$ | 0.25 | 0.33 | 0.5 |
|---|---|---|---|---|
| | 30ep | 76.04±0.03 | 76.84±0.17 | **77.93**±0.15 |
| | 60ep | 81.75±0.02 | 81.94±0.14 | **82.14**±0.13 |
| $\beta = 3$ | 90ep | 83.10±0.06 | **83.17**±0.18 | 82.99±0.16 |
| | 150ep | 83.98±0.21 | **83.99**±0.07 | 83.41±0.13 |
| | 300ep | **84.37**±0.04 | 84.04±0.07 | 83.41±0.11 |
| | 30ep | 75.03±0.18 | 76.46±0.21 | **77.72**±0.23 |
| | 60ep | 81.05±0.04 | 81.66±0.20 | **82.20**±0.13 |
| $\beta = 5$ | 90ep | 82.78±0.16 | 82.99±0.12 | **83.06**±0.14 |
| | 150ep | **83.92**±0.13 | 83.79±0.04 | 83.40±0.13 |
| | 300ep | **84.13**±0.14 | 84.02±0.08 | 83.35±0.15 |

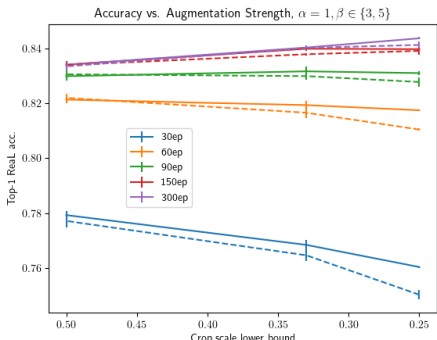

Figure 17: Top-1 val. accuracy (ReaL), Beta crop with $\alpha = 1$. Solid line: $\beta = 3$, Dashed line: $\beta = 5$

| | $a_{min}$ | 0.05 | 0.1 |
|---|---|---|---|
| | 30ep | 77.28±0.11 | **77.67**±0.18 |
| | 60ep | 82.28±0.16 | **82.34**±0.10 |
| $\alpha = 2$ | 90ep | 83.60±0.27 | **83.61**±0.01 |
| | 150ep | **84.43**±0.08 | 84.20±0.15 |
| | 300ep | **84.67**±0.21 | 84.41±0.04 |
| | 30ep | 77.79±0.17 | **77.86**±0.18 |
| | 60ep | **82.35**±0.18 | 82.22±0.14 |
| $\alpha = 3$ | 90ep | 83.37±0.02 | **83.44**±0.09 |
| | 150ep | **84.04**±0.14 | 83.91±0.15 |
| | 300ep | **84.12**±0.15 | 83.88±0.13 |
| | 30ep | 78.00±0.16 | **78.17**±0.21 |
| | 60ep | 82.01±0.20 | **82.09**±0.01 |
| $\alpha = 5$ | 90ep | **83.04**±0.09 | 82.85±0.11 |
| | 150ep | **83.48**±0.16 | 83.32±0.12 |
| | 300ep | **83.30**±0.24 | 83.19±0.13 |

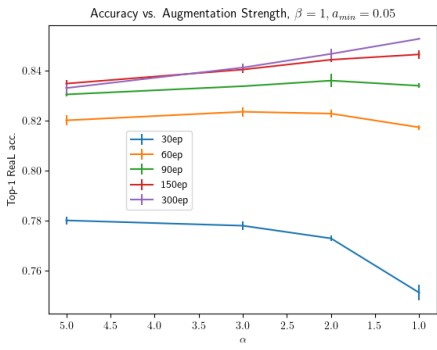

Figure 18: Top-1 val. accuracy (ReaL), Beta crop with $\beta = 1$. Top-1 val. accuracy vs. $\alpha$, $a_{min} = 0.05$ is plotted on the right. For $\alpha = \beta = 1$, we reuse the data points of the PyTorch Inception crop experiments.

| $\alpha$ | 1.149 | 1.287 | 1.551 | 1.818 | 2.099 | 2.403 |
|---|---|---|---|---|---|---|
| 30ep | 67.47±0.18 | 67.92±0.20 | 68.67±0.15 | 69.22±0.25 | 69.61±0.19 | **69.87**±0.04 |
| 60ep | 74.59±0.13 | 74.79±0.14 | 75.24±0.17 | 75.32±0.12 | 75.48±0.15 | **75.51**±0.04 |
| 90ep | 76.89±0.04 | 76.99±0.23 | **77.39**±0.11 | 77.25±0.15 | 77.15±0.14 | 77.29±0.21 |
| 150ep | 78.63±0.21 | 78.49±0.07 | **78.73**±0.06 | 78.59±0.24 | 78.63±0.09 | 78.59±0.12 |
| 300ep | 79.57±0.10 | **79.60**±0.11 | 79.42±0.10 | 79.27±0.15 | 78.95±0.23 | 79.04±0.12 |

Table 15: Top-1 val. accuracy of Beta crop, matching strength (original label). As the training budget increases from 30 epochs to 300 epochs, the optimal $\alpha$ decreases (stronger augmentation). $\alpha = 1.149$ may be an exception possibly due to too many crops near zero area. In future iterations we may want small but nonzero $a_{\min}$ (say $0 < a_{\min} \leq 0.025$) instead while keeping the same 5th percentile crop scale.

| $\alpha$ | 1.149 | 1.287 | 1.551 | 1.818 | 2.099 | 2.403 |
|---|---|---|---|---|---|---|
| 30ep | 75.16±0.15 | 75.58±0.17 | 76.44±0.12 | 76.97±0.23 | 77.30±0.08 | **77.58**±0.10 |
| 60ep | 81.72±0.05 | 81.85±0.06 | 82.20±0.10 | 82.17±0.16 | **82.35**±0.15 | 82.35±0.05 |
| 90ep | 83.48±0.04 | 83.56±0.11 | **83.72**±0.09 | 83.59±0.06 | 83.54±0.17 | 83.72±0.15 |
| 150ep | **84.72**±0.16 | 84.48±0.08 | 84.63±0.03 | 84.48±0.10 | 84.41±0.08 | 84.42±0.06 |
| 300ep | **85.15**±0.16 | 85.14±0.03 | 84.90±0.10 | 84.77±0.16 | 84.58±0.11 | 84.55±0.14 |

Table 16: Rop-1 val. accuracy of Beta crop, matching strength (ReaL). As the training budget increases from 30 epochs to 300 epochs, the optimal $\alpha$ decreases (stronger augmentation).

# H    More plots for gradient, Grad-CAM, and relevance entropy

## H.1    Gradient entropy

Figure 19 shows some sample saliency maps given by the rectified gradient. Plotting the mean gradient entropy against the top-1 validation accuracy using the multi-label ReaL does not change the whole picture: The mean gradient entropy is still negatively correlated with the training budget (fig. 20) as the distribution of $S_{\mathrm{grad}}$ for the 50000 validation set images shifts to the left (fig. 21).

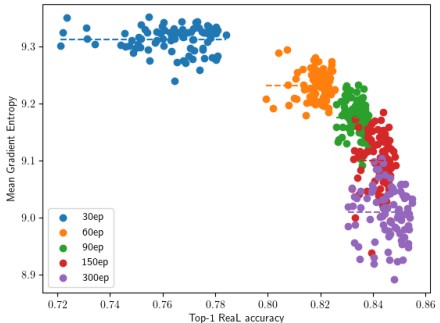

Figure 20: Mean gradient entropy $\overline{S_{\mathrm{grad}}}$ defined in eq. (2) vs. top-1 val. accuracy (ReaL) for all 450 Inception crop and Beta crop experiments with AdamW optimizer, color-coded by $a_{\min}$.

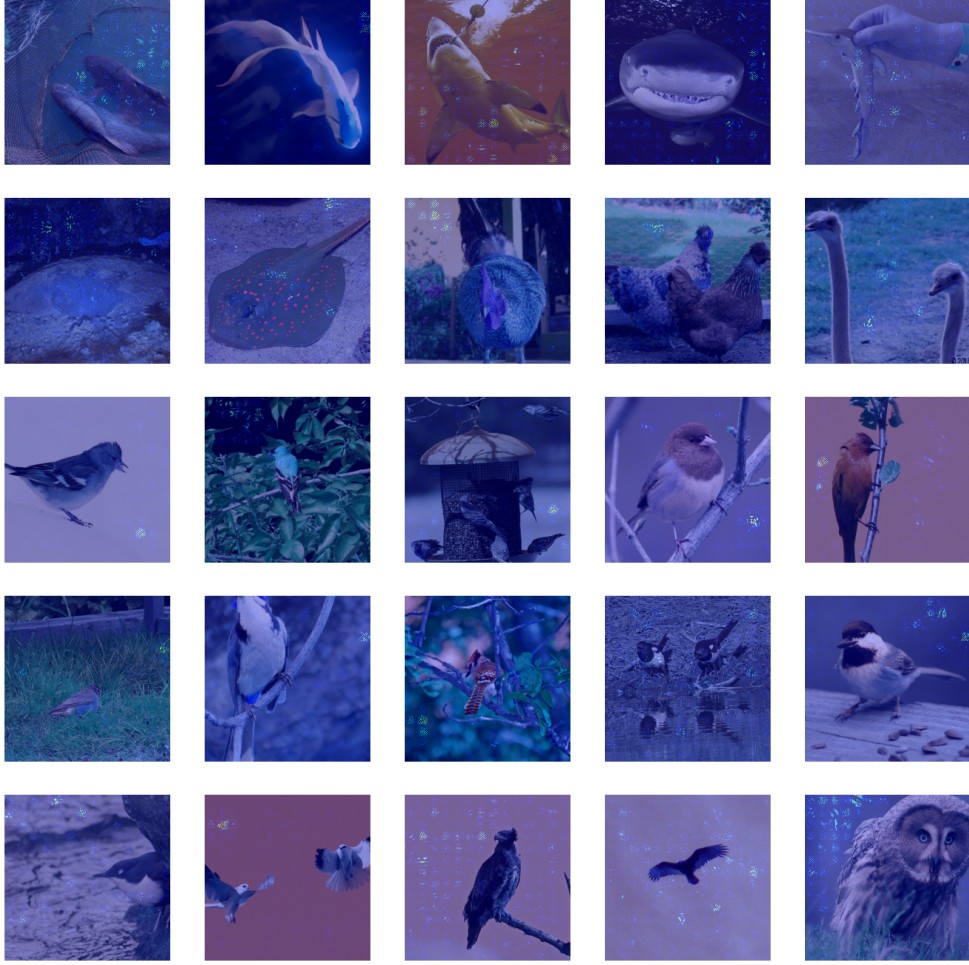

Figure 19: Sample saliency maps given by the rectified gradient for the 1st images of the first 25 classes $c \in [0..24]$ in torchvision.datasets.ImageNet validation split. We use a model trained with 300ep training budget and the TensorFlow Inception crop with $a_{\min} = 0.05$. Though conceptually the simplest, the saliency maps given by the rectified gradient are the least interpretable of the 3.

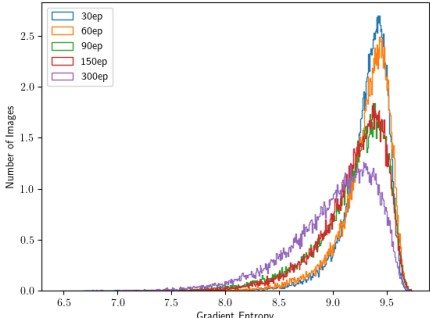 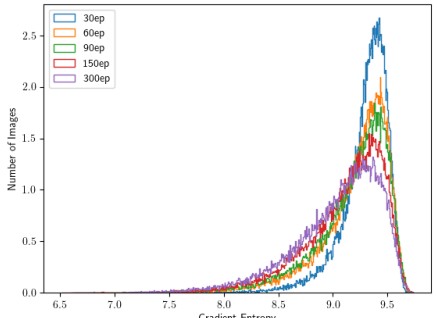

Figure 21: Distribution of the gradient entropy $S_{\text{grad}}$ defined in eq. (2) for the 50000 validation set images given by models trained with the TensorFlow Inception crop (left) and PyTorch Inception crop (right) with $a_{\min} = 0.05$, color-coded by the training budget.

## H.2 Grad-CAM entropy

Figure 22 shows some sample saliency maps given by Grad-CAM of the last layer and fig. 23 shows the distribution of the Grad-CAM entropy $S_{\text{grad}-\text{cam}}$ for the 50000 validation set images with various training budgets while keeping $a_{\min} = 0.05$ fixed.

## H.3 Relevance entropy

Figure 24 shows some sample saliency maps given by LRP of the last layer and fig. 25 shows the distribution of the relevance entropy $S_{\text{rel}}$ for the 50000 validation set images with various training budgets while keeping $a_{\min} = 0.05$ fixed. Compared to the gradient entropy counterpart (fig. 21) the sparsification at the last layer is more modest, but the trend remains: The higher the training budget, the sparser the attention becomes.

# I  Scion experiment setup and WD sweep

The constrained variant of Scion (Pethick et al., 2025) can be considered a collection of optimizers with the following unified update rules. Given loss function $f_t$ at time $t$, layer $l$ and layer weight $\boldsymbol{\theta}_{t,l}$ at time $t-1$, the choice of linear minimization oracle $\text{lmo}_l$, momentum $\alpha$, learning rate $\gamma$, and radius $\rho_l$:

$$\boldsymbol{g}_{t,l} \leftarrow \nabla_{\theta_l} f_t(\boldsymbol{\theta}_{t-1,l}, \zeta_t)$$
$$\boldsymbol{m}_{t,l} \leftarrow (1-\alpha)\boldsymbol{m}_{t-1,l} + \alpha\boldsymbol{g}_{t,l}$$
$$\boldsymbol{\theta}_{t,l} \leftarrow (1-\gamma)\boldsymbol{\theta}_{t-1,l} + \gamma\rho_l \,\text{lmo}_l(\boldsymbol{m}_{t,l})$$

Table 17 lists the lmos and the norms from which they are derived that we use in our experiments. Conceptually, we choose the norms of the layers based on the shape of the weight and their functions in the model, and lmos are the updates with unit norms in the direction of the steepest descent.

Although equivalent up to reparameterization, it is difficult to perform a WD sweep or the equivalent in the original formulation of Scion. We therefore reformulate constrained Scion in terms of independent weight decay coefficient $\eta = \gamma$, layer-wise learning rate $\gamma_l = \gamma\rho_l$, and layer-wise weight decay coefficient $\lambda_l = \frac{1}{\rho_l}$. The update rules then become

$$\boldsymbol{g}_{t,l} \leftarrow \nabla_{\theta_l} f_t(\boldsymbol{\theta}_{t-1,l}, \zeta_t)$$
$$\boldsymbol{m}_{t,l} \leftarrow (1-\alpha)\boldsymbol{m}_{t-1,l} + \alpha\boldsymbol{g}_{t,l}$$
$$\boldsymbol{\theta}_{t,l} \leftarrow (1-\eta)\boldsymbol{\theta}_{t-1,l} + \gamma_l \,\text{lmo}_l(\boldsymbol{m}_{t,l})$$
$$= \boldsymbol{\theta}_{t-1,l} + \gamma_l \left(-\lambda_l \boldsymbol{\theta}_{t-1,l} + \text{lmo}_l(\boldsymbol{m}_{t,l})\right)$$

We then adopt the following modifications from Pethick et al. (2025) for our Simple ViT-S/16:

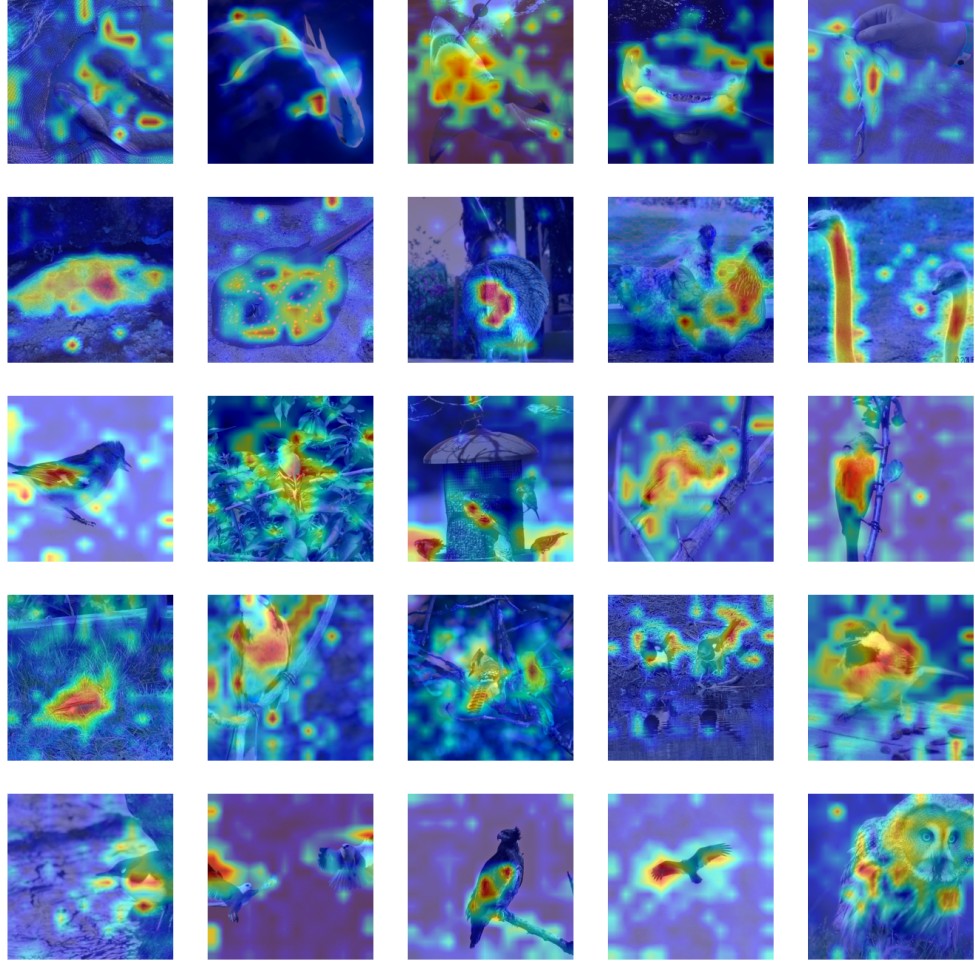

Figure 22: Sample saliency maps given by Grad-CAM at the last layer for the 1st images of the first 25 classes $c \in [0 . . 24]$ in `torchvision.datasets.ImageNet` validation split. We use a model trained with 300ep training budget and the TensorFlow Inception crop with $a_{\min} = 0.05$.

Table 17: Norms and the associated lmos as normalized in our experiments. Sign and Spectral assume matrix weight $\boldsymbol{\theta}_l = \boldsymbol{A} \in \mathbb{R}^{d_{\text{out}} \times d_{\text{in}}}$ while Bias assumes vector weight $\boldsymbol{\theta}_l = \boldsymbol{b}_\ell \in \mathbb{R}^{d_{\text{out}}}$. $\boldsymbol{U}\boldsymbol{V}^\top$ refers to the reduced SVD of the input matrix with unitary matrices $\boldsymbol{U}$ and $\boldsymbol{V}^\top$ from the full SVD $\boldsymbol{A} = \boldsymbol{U}\text{diag}(\boldsymbol{\sigma})\boldsymbol{V}^\top$ while $\|\boldsymbol{A}\|_{\mathcal{S}_\infty} = \max(\boldsymbol{\sigma})$ is the spectral norm of the matrix.

|  | Sign | Spectral | Bias |
|---|---|---|---|
| **Norm** | $d_{\text{in}} \max_{i,j} |A_{i,j}|$ | $\sqrt{\frac{d_{\text{in}}}{d_{\text{out}}}} \|\boldsymbol{A}\|_{\mathcal{S}_\infty}$ | RMS |
| **LMO** | $\boldsymbol{A} \mapsto -\frac{\text{sign}(\boldsymbol{A})}{d_{\text{in}}}$ | $\boldsymbol{A} \mapsto -\sqrt{\frac{d_{\text{out}}}{d_{\text{in}}}}\boldsymbol{U}\boldsymbol{V}^\top$ | $\boldsymbol{b}_\ell \mapsto -\frac{\boldsymbol{b}_\ell}{\|\boldsymbol{b}_\ell\|_{\text{RMS}}}$ |

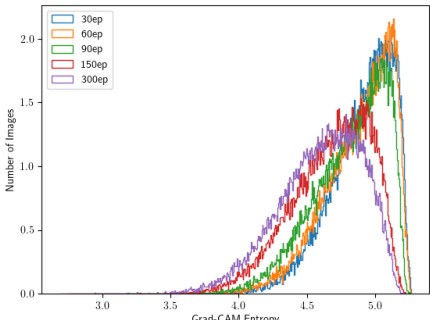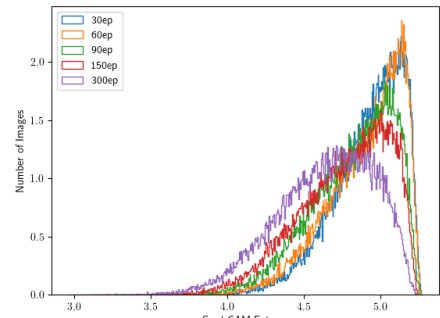

Figure 23: Distribution of the Grad-CAM entropy $S_{\mathrm{grad-cam}}$ for the 50000 validation set images given by models trained with the TensorFlow Inception crop (left) and PyTorch Inception crop (right) with $a_{\min} = 0.05$, color-coded by the training budget.

1. Scale the GELU activation function as $\sqrt{2}$GELU to preserve variance.

2. Replace LayerNorm with RMSNorm.

We also keep its batch size 4096, cosine learning rate decay with no warm-up, choices of lmos (Spectral for the input patchifier and hidden layers, Sign for the output layer, and Bias for the biases) and maximum learning rates ($\gamma_L = \gamma\rho_L = 0.0004 \times 500 = 0.2$ for the output Sign layer and $\gamma_l = \gamma\rho_l = 0.0004 \times 25 = 0.01$ for the rest). The WD coefficient (converted to our formulation) from Pethick et al. (2025) turns out to be suboptimal so we end up doing our own sweep with $\lambda_l \in \{0.04, \mathbf{0.08}, 0.12, 0.16\}$ and $\lambda_L = 0.05\lambda_l$ for the output Sign layer. The best value $\lambda_l = 0.08$ (table 18) is used for the comparison in section 6.4. Including the optimized $a_{\min}$ experiments (table 18), we have a total of 72 models trained with Scion optimizer. Plotting their mean gradient entropy $\overline{S_{\mathrm{grad}}}$ defined in eq. (2) vs. top-1 val. accuracy, original label (left) and ReaL (right), for all 72 experiments with Scion optimizer, color-coded by $\lambda_l$ shows no correlation between $\overline{S_{\mathrm{grad}}}$ and WD (fig. 26).

| $\lambda_l$ | Original label | | | | ReaL | | | |
|---|---|---|---|---|---|---|---|---|
| | 0.04 | 0.08 | 0.12 | 0.16 | 0.04 | 0.08 | 0.12 | 0.16 |
| 30ep | 72.08±0.19 | **73.31**±0.09 | 73.24±0.21 | 72.50±0.22 | 79.46±0.18 | **80.83**±0.07 | 80.81±0.13 | 80.18±0.20 |
| 60ep | 76.98±0.09 | **77.44**±0.09 | 76.88±0.14 | 76.07±0.16 | 83.58±0.08 | **84.09**±0.10 | 83.75±0.15 | 83.26±0.18 |
| 90ep | 78.42±0.19 | **78.68**±0.09 | 78.32±0.04 | 77.31±0.12 | 84.51±0.08 | **84.94**±0.05 | 84.88±0.05 | 84.11±0.10 |
| 150ep | 79.27±0.08 | **79.65**±0.07 | 79.05±0.15 | 78.31±0.07 | 84.94±0.09 | **85.41**±0.05 | 85.20±0.14 | 84.69±0.08 |
| 300ep | 79.67±0.06 | **80.10**±0.14 | 79.98±0.10 | 78.64±0.32 | 85.04±0.10 | **85.66**±0.21 | 85.66±0.09 | 84.54±0.35 |

Table 18: Top-1 val. accuracy of the Scion optimizer WD sweep, original label (left) and ReaL (right).

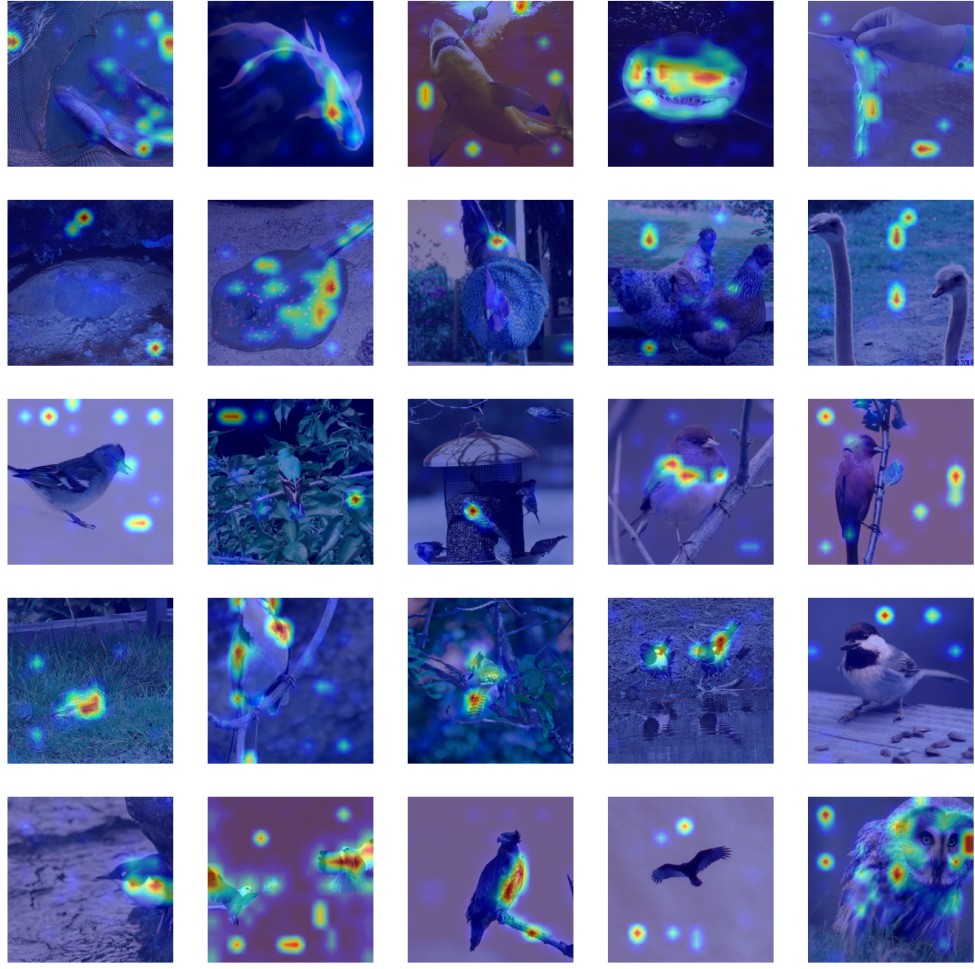

Figure 24: Sample saliency maps given by LRP at the last layer for the 1st images of the first 25 classes $c \in [0 .. 24]$ in `torchvision.datasets.ImageNet` validation split. We use a model trained with 300ep training budget and the TensorFlow Inception crop with $a_{\min} = 0.05$.

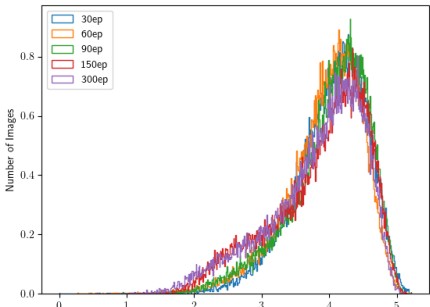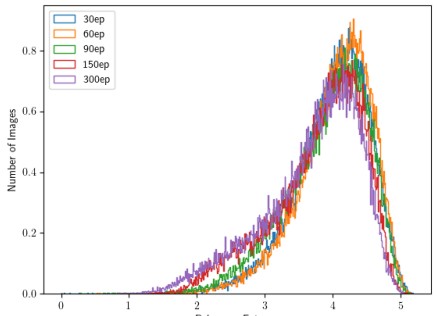

Figure 25: Distribution of the relevance entropy $S_{\mathrm{rel}}$ for the 50000 validation set images given by models trained with the TensorFlow Inception crop (left) and PyTorch Inception crop (right) with $a_{\min} = 0.05$, color-coded by the training budget.

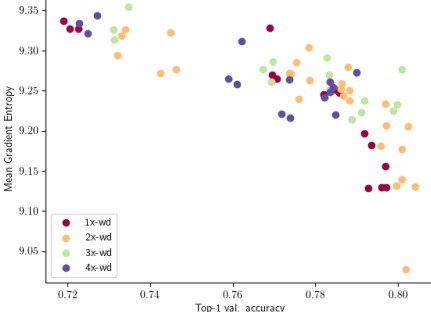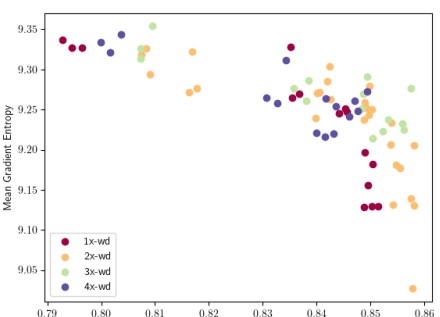

Figure 26: Mean gradient entropy $\overline{S_{\mathrm{grad}}}$ vs. top-1 val. accuracy, original label (left) and ReaL (right), for all 72 experiments with Scion optimizer, color-coded by $\lambda_l$.

