# OpenReview forum: "You May Be Running the Wrong Inception Crop"
_TMLR — Rejected by TMLR_

### Review · Reviewer_tMK2 · 2026-03-06

**Summary Of Contributions:**

This paper highlights a hidden implementation difference between the Pytorch and TF/JAX versions of the "inception crop" which is commonly used in ImageNet classifier training. The authors find that the TF implementation oversamples smaller areas compared to pytorch, leading to minor performance degradation. The authors then propose a more flexible sampling scheme based on a beta distribution, as well as explore the effects of cropping on saliency sparsity.

**Strengths**
- The implementation inconsistency pointed out is of some practical value to the community at large.
- There are some useful insights about ImageNet training and augmentation, such as the interaction between the minimum crop size and validation accuracy at different compute budgets.

**Weaknesses**

If I have made any mistakes in reading/understanding the paper, I welcome the authors to correct me.

1. The differences in empirical performance are quite minor across the board, especially for the best-performing models that have been trained for the longest. Comparing the bottom row of Table 1 the difference between TF and pytorch is at most ~0.4 percentage points and often less than 0.1.
1. The usefulness of the results seems to be tightly tied to ImageNet classification, which is an academic benchmark that is becoming increasingly divorced from the real-world application of deep learning to vision in the year 2026. The authors didn't provide any deeper *understanding* into the underlying relationship between the data (imagenet), task (classification), augmentation (cropping) and generalisation performance. They don't even remark that $a_{min}=0$ will result in some tiny crops that do not contain any pixels useful for discrimination. As such, I feel like there is a lack of generalisable takeaways that would be useful for tasks closer to real world practice such as semantic segmentation/object detection, large-scale image encoder training like CLIP or generative model training such as DiT. Noteably for these tasks random cropping is either much less aggressive or simply not used (DiT uses a simple square centre crop).
1. The paper does not consider at all how cropping may (or may not) interact with other augmentations approaches that are present such as mixup.
1. The proposed beta crop adds hyperparameter complexity without materialising much accuracy benefit (and is worse for 300 epoch training compared to pytorch).
1. It is unclear what the reader is meant to takeaway from the saliency results (although the reviewer is not that familiar with explainable/interpretable AI).
1. The presentation of the paper could be improved; tables could use booktabs, algorithms would benefit from clarifying comments, and the text in the figures is far too small.
1. In Sec 3.1 the claim that the PDF for area is proportional to 1/sqrt(area) is not clearly derived from algorithm 1.

**Audience:**

No

**Audience Explanation:**

The contributions, although clear, have such narrow implications (a few tenths of a percentage point on ImageNet classification), that I find it hard to justify publishing them at TMLR, however, I will defer to the action editor if they believe otherwise.

**Claims And Evidence:**

Yes

**Claims Explanation:**

The results presented seem reasonable and the experiments are extensive (although they are narrowly focused on ImageNet and ViT).

**Requested Changes:**

I don't believe the authors can revise the current manuscript in a way that would convince me it would be suitable for publication at TMLR. Perhaps a *wider* exploration of random cropping across more applications, or a *deeper* investigation into the relationship between generalisation, data and cropping parameters would take the research in a direction of greater value and they could resubmit in the future. As it stands I feel like the main takeaways of the paper would be more suitable for a blogpost.

---

> ### Author Response · Authors · 2026-03-06
> **Correct the Record on Weakness 2**
>
> We thank reviewer tMK2 for their thoughtful review. Just to correct the record on weakness 2: We did point out that matching-strength Beta crop with $\alpha = 1.149$ may be underperforming due to too many crops near zero area in the caption of Table 3 on page 9. We chose $a_{\min}=0$ regardless based on the rationale that PDF of beta distribution $x \sim \mathcal{B}(\alpha > 1,\beta=1)$ already vanishes at $x=0$ and increasingly rapidly with larger $\alpha$. As stated, we suggested small but nonzero $a_{\min}$ for future iterations since the results indicate that it might not vanish fast enough.
>
> We also can't quite agree with the characterization that the reported performance differences are minor and that ImageNet classification is increasingly divorced from the real-world application. E.g. the last published version of MobileNet (MobileNetV4, ECCV 2024) uses Dynamic Dataset Mixing that includes ImageNet-1k and a subset of JFT-300M, both augmented with Inception Crop, and the performance gain is also a few tenths of a percentage point on ImageNet classification (Table 8, [1]). They didn't specify the $a_{\min}$ value they use but since they trained for at least 400 epochs and used TensorFlow, it is likely the TensorFlow default $a_{\min} = 0.05$ or similar and their results are most likely impacted by the implementation bug we reported.
>
> For weakness 7, Algorithm 1 samples the height of the crop uniformly in the interval $[h_{\min}, h_{\max}]$. Since $h \propto$ the square root of the crop area $\sqrt{A}$ once we fix the aspect ratio, CDF of the crop area follows $\Pr(a < A) \propto \sqrt{A} - C$. PDF $f(A) \propto \frac{1}{\sqrt{A}}$ then follows by taking derivative w.r.t. $a$.
>
> 1. Qin, Danfeng, et al. "MobileNetV4: Universal models for the mobile ecosystem." European conference on computer vision. Cham: Springer Nature Switzerland, 2024.

---

> ### Comment · Reviewer_tMK2 · 2026-04-14
>
> Hi, this is just to let the authors know that I will get back to them in light of their above response as well as the other reviews later this week (I am currently a bit busy with other work).

---

> > ### Comment · Reviewer_tMK2 · 2026-04-21
> >
> > Apologies for the delay, here is my further response to the authors.
> >
> > - **ImageNet classification**: although it is true that MobileNetV4 uses, as part of its training data, ImageNet, it is also the case that the application of the architecture is generally on other downstream tasks (e.g. as a backbone for edge object detection), rather than ImageNet classification. I think the submission would be stronger if it had investigated such tasks (as already discussed).
> > - **Area distribution**: I'm not sure if the 1/sqrt(A) result is correct. Although height is sampled uniformly, so is the aspect ratio, making the area the product of the square of one uniform RV and another uniform RV $A=h^2r$. Considering figure 1, it is also clear that 1/sqrt(A) is not a good fit for the empirical distribution.
> > - **Final thoughts**: After considering the authors' response as well as their discussions with the other reviewers, my general position on the paper remains unchanged. I do not believe the contribution, in terms of the advancement of knowledge for the research community, is sufficient for publication at this venue. That is not to say that I believe this paper is without value, but rather it is not (yet) suitable for TMLR. I think the current contents of the paper would be most suitably disseminated via a blog post. If the authors wish to improve the paper for future submission, I would suggest either a *deeper* (more insight into the interaction between augmentation, length of training, and generalisation) or *broader* (wider experimental scope, e.g. image generation, object detection etc.) investigation. I hope the authors find this feedback helpful.

---

> > > ### Author Response · Authors · 2026-04-21
> > >
> > > * Re: Area distribution:
> > >
> > > To be exact, it is the conditional PDF of the crop area $A$ given the aspect ratio $r$ that is proportional to $\frac{1}{\sqrt{A}}$ within the support, i.e. $f(A | r) \propto \frac{1}{\sqrt{A}}$ where $f(A | r) \neq 0$. The marginal PDF $f(A)$ is truncated at larger crop area $A$ when some aspect ratios become inadmissible as described in the caption of figure 1. For a $256 \times 512$ image with parameters $r_\min = \frac{3}{4}$ and $r_\max = \frac{4}{3}$, only $A \le 256^2 \times \frac{3}{4} = 49152$ can be sampled with all aspect ratios $r_\min \le r \le r_\max$, resulting in the apparent deviation. In fact, the same applies to the correct Inception crop implementation (equivalent to Beta crop with $\alpha = \beta = 1$): If $a_\max = 1$, $f(A | r)$ can be constant within the support but not the marginal PDF $f(A)$. See the crop area distribution of Beta crop with $\alpha = \beta = 1$ in figure 3.
> > >
> > > * Re: Final thoughts:
> > >
> > > We respect the stance of Reviewer tMK2. We do want to remind reviewers that the second acceptance criteria of TMLR is "Would some individuals in TMLR's audience be interested in the findings of this paper?". In this context it seems odd to advocate that the current content of the paper should be a blog post instead.

---

### Review · Reviewer_PVtk · 2026-03-07

**Summary Of Contributions:**

The paper investigates a commonly used random-crop augmentation strategy and highlights a discrepancy between its implementations in two major ecosystems, PyTorch (PT) and TensorFlow/JAX (TF). The authors show that this discrepancy can have a (relatively minor) effect on supervised training with ViT-S. The paper also includes discussion for an alternative BetaCrop strategy.

The paper declares three central "findings" as key contributions;
1. Augmentation strength depends on the training budget.
2. Tail of distribution for crop scale determines the augmentation strength.
3. Models trained with higher training budgets exhibit sparser saliency.

Of these, only (2) and (3) seem to be actual contributions. Claim (1) is not new in substance, and is already broadly supported by prior work, including Steiner et al. (2022), which showed that augmentation and regularization interact strongly with training duration. While the paper partially acknowledges this context, it still elects to present this result as a standalone finding. Furthermore, the evidence for claims (2), (3) is somewhat lacking, discussed more in claims and evidence below.

After these findings, the paper goes on to propose BetaCrop which reads as a contribution, confusingly while later stating this as previously proposed by Bouchacourt et al. (2021). As a result, the main contribution is less the introduction of BetaCrop itself than the particular empirical analysis carried out, generally constrained to varying the $\alpha$ parametrization during sampling.

More broadly, the contribution profile of the paper is somewhat uneven. The most robust empirical observation (that augmentation strength interacts with training budget) is already well established in prior work, while the more novel claims (regarding the role of crop-scale tails and saliency sparsification) are supported only within a somewhat narrow experimental setting (small capacity, focus on limited epochs).

**Additional Comments:**

We provide a short summary of the scope of the review for reference.

**Strengths**

- The paper revisits one of the most commonly used augmentation strategies in modern vision pipelines.
- It identifies and analyzes an implementation discrepancy between two major frameworks (PyTorch and TensorFlow/JAX).
- The experimental design is reasonably thorough within the stated setup, exploring multiple training budgets and analysis directions.

**Weaknesses**

- Several contributions appear overstated, and some of the key findings are already established in prior work.
- The strongest empirical observation (interaction between training budget and augmentation strength) is already known in the literature, while the more novel claims are supported only within a narrow experimental setting.
- The paper omits relevant cropping strategies previously explored for ViT training (e.g., SRC, RICAP).
- The reported empirical differences between the “faulty” and “correct” implementations are often small and not clearly conclusive.
- The experimental scope (ViT-S, supervised ImageNet training up to 300 epochs) limits the ability to generalize the conclusions to current extended training practices.
- The presentation and formatting of results make empirical comparisons unnecessarily difficult to follow.
- Some parts of the study appear orthogonal to the central question, particularly the saliency analysis and the optimizer ablation with limited baselines.

**Audience:**

No

**Audience Explanation:**

From the abstract and claims, this reviewer was genuinely interested in seeing what the community has missed out on by focusing exclusively on a specific cropping strategy, that while proven, has resisted change for years. However, the results of the paper are unclear at best, and lacking in evidence and baselines. Seeing as the community is pivoting towards ViTs with larger capacity (base or large) with higher training budgets (as advocated by modern best practices); the takeaway that most practitioners are likely **not** running the wrong inception crop.

While the topic is interesting due to its importance in nearly all training pipelines, the unclear declaration of findings (finding 1 and BetaCrop are already known in the literature), limited scope (ViT-S and focus on low epochs), coupled with weak or inconclusive evidence, I do not find the findings interesting enough to warrant publication in TMLR.

**Broader Impact Concerns:**

No broader impact statement seems necessary, the paper focuses on image augmentation via cropping strategies.

**Claims And Evidence:**

No

**Claims Explanation:**

The paper elevates an implementation discrepancy into an ecosystem-level concern, but in the training regime most relevant for standard ViT pretraining, the observed differences are inconsistent and often practically negligible, (neither PT nor TF are universally better). While the authors are somewhat exhaustive within the stated framework, there is reason to be skeptical that a ViT-S with maximum 300 epochs in a supervised setting is enough to cover modern training regimes. DEiT3 by Touvron et al. (2022) advocates for 400 epochs and DINO by Caron et al. (2021) use 800 epochs for ViT-S. At the same time, the paper’s strongest numerical gaps appear in the short-budget settings, especially 30 and 60 epochs, where TF underperforms PT more noticeably. But that is exactly where the relevance becomes murkier if one is thinking about standard ViT training practice rather than low-budget ablations. As such, it seems the paper only measures meaningful differences for exceedingly short training regimes, reducing the impact of the observed discrepancy between implementations. These observations on scope also applies to the study on BetaCrop, see below.

Moreover, there is a clear omission of baselines for the study of cropping strategies for ViTs. DEiT3 by Touvron et al. (2022) includes an alternative cropping strategy (SRC) that is found to be stronger for supervised training than Inception Crop. Other examples include RICAP by Takahashi et al. 2019. These seem within the scope of the experimental setup of the paper, so their omissions are concerning, and the paper does not seem to look to include alternatives except BetaCrop in their study.

The support for claims (2) and (3) does not fully match the way they are framed. For claim (2), the argument relies on a particular empirical proxy for augmentation strength rather than a more general analysis. The authors show that a lower-tail statistic tracks performance reasonably well in their sweeps, but not that the lower tail is the decisive causal quantity. The claim

For claim (3), the saliency result appears primarily observational, with limited theoretical or practical integration into the rest of the paper, so the analysis is difficult to interpret in the context of the paper’s main claims. The results rely on specific interpretability methods (GradCAM and LRP), which are known to produce method-dependent and sometimes unstable explanations. Without comparing multiple interpretability approaches or providing a stronger mechanistic link to the augmentation findings, it is unclear what role these observations play in supporting the central argument of the paper. As presented, the saliency section appears largely observational and orthogonal to the main investigation of crop distributions.

As for BetaCrop, the variant proposed does not seem to provide a central advantage in comparison to others. There are some gains, sure, and BetaCrop is a neater mathematical generalization, which also aligns with sampling in CutMix by Yun et al. (2019). However, this is not a  uniquely novel contribution. Coupled with a lack of quantitative support for the proposed variant, the results seem much too weak for publication.

Issues with reporting (clarity of tables and discussion, see below) makes the evidence for the claims unclear. Tables 1,2, and 3 are formatted in a manner that makes direct comparison exceedingly hard on the reader, and the comparison is made across multiple tables on different pages. After parsing these results, this reviewer sees little to support the urgency of the claims made by the authors.

Lastly, the role of the Scion optimizer in this study is unclear. If the intention is to demonstrate that the observed effects are robust across optimizers, the evidence is insufficient: only AdamW and Scion are considered. AdamW is still the standard, and is well chosen as a baseline. However, Scion has limited adoption in vision compared to more widely used alternatives such as standard SGD with momentum or LAMB (e.g., in DEiT3), and recent work on low-budget training has explored optimizers such as Muon. Without a clear motivation for including Scion specifically, this section adds another variable to the experiments without resolving a meaningful ambiguity.

**Requested Changes:**

## Major changes

These concern the central claims of the paper and would require additional experimental evidence to support them.

**1. Broaden experimental scope beyond supervised ViT-S training**

The current experiments are restricted to supervised ImageNet-1k training with ViT-S up to 300 epochs. Since the conclusions are framed broadly, it would strengthen the claims substantially to include additional regimes, such as:

- larger models (e.g., ViT-B)
- longer training schedules
- self-supervised training settings
- fine-tuning scenarios

Without evidence beyond this narrow setup, it is difficult to assess whether the reported observations generalize.

**2. Include comparisons against additional cropping strategies**

The experimental study focuses almost exclusively on Inception Crop variants and BetaCrop. However, several other cropping-based augmentation strategies have been explored in the literature and are within the scope of the paper’s experimental setup, including:

- SRC from DEiT3 (Touvron et al., 2022)
- RICAP (Takahashi et al., 2019)

Including such baselines would provide a clearer picture of whether BetaCrop represents a meaningful improvement over existing alternatives.

**3. Clarify the novelty and role of BetaCrop**

The manuscript presents BetaCrop as a proposed method, but later acknowledges that related parameterizations were previously explored by Bouchacourt et al. (2021). The contribution would benefit from a clearer statement of what is novel in this work:

- Is the novelty the parameterization itself?
- The empirical analysis?
- The connection to crop distribution tails?

Clarifying this would help avoid confusion about the contribution.

**4. Clearly acknowledge and contextualize previous works**

Finding (1) is well known, and not an independent contribution. As a claim, it needs to be posed as a clarification and validation of previous findings explicitly in the introduction of the stated claims and contributions. Since this behaviour has already been documented in prior work, it would be prudent to explicitly frame this result as confirmation or clarification of existing findings rather than presenting it as an independent contribution.

**5. Clarify or remove orthogonal claims**

The findings on saliency and ablation of Scion comes across as unmotivated. To warrant its inclusion, a convincing argument must be made towards *why these experiments matter for cropping*. The current language is unclear and informal, merely stating that the authors hypothesise without elaboration. The manuscript should clarify how these experiments contribute to the main argument of the paper. If a clear connection cannot be established, the authors may consider simplifying the paper by removing or substantially reducing these sections.

## Minor changes / edits / comments

The presentation of the paper leaves many things to be desired. As mentioned in in claims and evidence, the experimental results are formatted in a way that makes it difficult to follow the arguments made in the paper, in particular, parsing Tables 1 and 2 seems to give very minor differences in general between the two implementations (PT vs TF).

Figures 1 and 4 is needlessly small for filling a whole page horizontally, and could easily be expanded for better readability. Algorithms 1 and 2 are included as floating figures or tables, and are missing captions.

Moreover, the submitted manuscript makes use of awkward wording that makes it very unclear what the authors actually mean.

> [BetaCrop] fixes $\alpha=1$ while varying $\beta$ [...] which do not cover more modest deviation from uniform distribution ...

As written, this sentence does not make sense. What does it mean to *"not cover more modest deviation"*?

Section 6.1. states plainly "Here are the results..." as an information dump mixing interpretation and discussion about parametrization. This is very poor practice in presentation of results, and adds unneccessary effort to the reader in parsing the key results. Importantly, it makes it very difficult to disambiguate any actual insights made by the authors regarding their results.

The manuscript occasionally mixes *crop area* vs. *crop scale* without reminding the reader what is meant. Since scale corresponds to area fraction, clarity on this terminology matters.

BetaCrop parameter sweeps appear without much explanation for why certain α values are chosen. The text jumps straight to:

> “We then sweep the parameter space…”

without motivating the parameter grid.

Also, examples in the manuscript include phrases like:

- “we later learn of…”
- “we then sweep…”
- “we can see…”

These read like an informal / internal research log rather than an academic paper.

---

> ### Author Response · Authors · 2026-03-10
> **Threaded reply (1/3)**
>
> We thank reviewer PVtk for their extensive review. Below is our threaded reply.
>
> # Summary Of Contributions:
>
> > Of these, only (2) and (3) seem to be actual contributions. Claim (1) is not new in substance, and is already broadly supported by prior work, including Steiner et al. (2022)
>
> By "Higher training budget requires stronger augmentation", we refer to the specific case of optimal Inception / Beta Crop vs. training budget and this should be clear given the context. It lends support to the general case but by no means do we claim it as new contribution. We already cited Steiner et al. (2022) and we would also like to see evidence of broad support by prior work and add them to the citations. Right now when we google "augmentation strength and training budgets" in an incognito browser window, Steiner et al. (2022) is the 1st result and our submission is the 4th.
>
> > After these findings, the paper goes on to propose BetaCrop which reads as a contribution, confusingly while later stating this as previously proposed by Bouchacourt et al. (2021).
>
> Bouchacourt et al. (2021) tested Beta crop but prematurely concluded that it underperforms Inception crop with nonuniform crop scale distribution. We carry out more extensive comparison and find it competitive when we tune its augmentation strength properly based on conclusion 2. We already stated in the abstract that our proposal is based on conclusion 2 but we are open to adjust the wording to be clearer, e.g. "We propose Beta crop with tuned augmentation strength based on 2".
>
> # Explanation for the answer whether the claims are supported:
>
> Overall we are rather confused by the explanation of reviewer PVtk. The extent to which the result generalizes and covers modern training regimes at most concerns whether the audience would be interested, not whether the claims are supported. Neither does whether the result is observational or uniquely novel. Can reviewer PVtk clarify where support of the claims is lacking? Nevertheless, We reply to some of the points raised below.
>
> > The paper elevates an implementation discrepancy into an ecosystem-level concern, but in the training regime most relevant for standard ViT pretraining, the observed differences are inconsistent and often practically negligible, (neither PT nor TF are universally better)
>
> That's only true if we disregard the implementation discrepancy and compare PT and TF Inception crop at the same $a_{\min}$. If we compare them at the optimally tuned augmentation strength or at the same augmentation strength measured by the 5th percentile crop scale, PT Inception crop is always either within the error bar or better.
>
> > (...) there is reason to be skeptical that a ViT-S with maximum 300 epochs in a supervised setting is enough to cover modern training regimes. DEiT3 by Touvron et al. (2022) advocates for 400 epochs and DINO by Caron et al. (2021) use 800 epochs for ViT-S.
>
> DINO is a self-supervised training method so the number of epochs by Caron et al. (2021) may not be informative for supervised training. Meanwhile, Steiner et al. (2022) also trains ViT on ImageNet-1k for 300 epochs.
>
> > Moreover, there is a clear omission of baselines for the study of cropping strategies for ViTs. DEiT3 by Touvron et al. (2022) includes an alternative cropping strategy (SRC) that is found to be stronger for supervised training than Inception Crop. Other examples include RICAP by Takahashi et al. 2019.
>
> SRC only outperforms PT Inception crop for ImageNet-21k pre-training and Touvron et al. (2022) give their reasoning why that is the case. RICAP is a mixing / cropping augmentation method more similar to CutMix. More importantly, neither is tunable in terms of crop scale or similar. The only hyperparameter of RICAP controls the degree of mixing instead. Therefore, their relevance to our setup and claims is limited.
>
> > For claim (2), the argument relies on a particular empirical proxy for augmentation strength rather than a more general analysis. The authors show that a lower-tail statistic tracks performance reasonably well in their sweeps, but not that the lower tail is the decisive causal quantity.
>
> Yes, lower-tail statistics such as the 5th percentile of the crop scale distribution is a good empirical proxy for augmentation strength. We are not claiming more than that.

---

> > ### Author Response · Authors · 2026-03-10
> > **Threaded reply (2/3)**
> >
> > Regarding claim (3), the saliency result:
> >
> > We give our reasoning why we investigate the saliency map of the models we trained in Sec. 6.3 on page 10: The crop scale determines the proportion of the image the model sees, so it is of interest whether changing the crop scale distribution changes the model saliency. In fact, Takahashi et al. (2019) did the same by using CAM to investigate how the model saliency is affected by training with RICAP. As for the concern that the results rely on specific interpretability methods, we measure the saliency sparsity in 3 ways (entropies of rectified gradient, GradCAM, and LRP) precisely to address this concern. If all results point in the same direction, the parsimonious explanation is that it is an inherent property of model training instead of any specific methods.
> >
> > > Lastly, the role of the Scion optimizer in this study is unclear. If the intention is to demonstrate that the observed effects are robust across optimizers, the evidence is insufficient: only AdamW and Scion are considered.
> >
> > Robustness w.r.t. optimizer is not all or nothing. Criticism of the choice of Scion due to limited adoption is particularly unwarranted for two reasons:
> >
> > 1. It substantially outperforms AdamW.
> > 2. The Spectral lmo update rule is identical to that of Muon, one of the more widely used alternatives listed.
> >
> > Given that Muon is typically used in combination with an auxiliary AdamW optimizer, in fact, testing Scion optimizer instead of Muon eliminates the possibility that the result depends on some model parameters still being trained with AdamW.
> >
> > # Explanation for the answer whether TMLR's audience be interested:
> >
> > > Seeing as the community is pivoting towards ViTs with larger capacity (base or large) with higher training budgets (as advocated by modern best practices); the takeaway that most practitioners are likely not running the wrong inception crop.
> >
> > The training budget at which the model's performance is the least sensitive to augmentation strength is actually 150 epochs, not 300 epochs. At 300 epochs the optimal $a_{\min}$ is $0.05$ for TF Inception crop and going lower leads to significantly degraded performance while lowering $a_{\min}$ to $0.025$ results in the same or improved performance for PT Inception crop. This is consistent with Conclusion 2: When measured by the 5th percentile crop scale, PT Inception crop with $a_{\min} = 0.025$ is in fact still slightly weaker augmentation than TF Inception crop with $a_{\min} = 0.05$ (Fig. 17 on page 23).
> >
> > # Requested Changes:
> >
> > ## Major changes
> >
> > 1. Broaden experimental scope beyond supervised ViT-S training
> >
> > Properly doing so with larger models or self-supervised training would make the submission far more extensive and require substantial compute resource while we are already at the regular submission limit of 12 pages. We are willing to offer running 400 epoch experiments given that reviewer PVtk is interested in the regime of higher training budgets but only if reviewer PVtk commits to recommend acceptance with their inclusion. Our predictions are straightforward but just to preregister them here: We predict that the optimal augmentation strength for 400 epochs will be the same or stronger than that of 300 epochs, the PT Inception crop will continue to perform the same or better than TF Inception crop at the same augmentation strength, and the saliency sparsity will either stay the same or increase when compared to the 300 epoch counterparts.
> >
> > 2. Include comparisons against additional cropping strategies
> >
> > As reasoned above, the relevance of SRC and RICAP to our setting is limited.
> >
> > 3. Clarify the novelty and role of BetaCrop
> >
> > This is fair. As explained and stated above, we are open to adjust the wording as either "We propose Beta crop with tuned augmentation strength based on 2", "We find that Beta crop, with augmentation strength tuned based on 2", or similar.
> >
> > 4. Clearly acknowledge and contextualize previous works
> >
> > We already cited Steiner et al. (2022) and are fully open to adding missing citations.
> >
> > 5. Clarify or remove orthogonal claims
> >
> > As explained neither the saliency sparsity nor the Scion result is orthogonal and there is precedence in investigating vision model saliency w.r.t. augmentation based on cropping (Takahashi et al. 2019). We are open to citing it and elaborating the connection.

---

> > > ### Author Response · Authors · 2026-03-10
> > > **Threaded reply (3/3)**
> > >
> > > ## Minor changes / edits / comments
> > >
> > > Overall we are open to editing such as making Figures 1 and 4 bigger, but issues such as the difficulty of comparing multiple tables across different pages is due to the inherent limitation that tables are 2-dimensional of limited size while we are presenting multidimensional data. If there are specific comparisons that reviewer PVtk is interested in we can create customized tables and add them to the appendix.
> > >
> > > Regarding Algorithms 1 and 2: They do have captions as titles of the algorithms, **sample_distorted_bounding_box()** and **RandomResizedCrop.get_params()** respectively. This is the intended way to use the algorithmic environment.
> > >
> > > > As written, this sentence does not make sense. What does it mean to "not cover more modest deviation"?
> > >
> > > We just meant that Bouchacourt et al. (2021) did not test Beta crop with crop scale distribution that deviates less from the uniform distribution, especially distributions with negative skew like in the matching-strength experiments. We will change the wording in the next revision.
> > >
> > > Regarding Section 6.1: This is also fair. We will trim the interpretation and discussion in Sec. 6.1 in the next revision.
> > >
> > > Regarding crop area vs. crop scale, "crop area" only appears 5 times in our submission. The first is in the footnote of the 1st page as the definition of crop scale and the rest are used in the captions and references of Fig. 2, 4, and 16. We used crop area for these figures since they are about crops of images of specific sizes and we find absolute sizes more intuitive in this context.
> > >
> > > Regarding parameter sweeps, we have been systematic but the exact grid points are inevitably somewhat arbitrary. One exception in our case is the matching-strength experiment, in which the value of $\alpha$ is determined solely by $a_{\min}$ of the current and prior sweeps.
> > >
> > > Regarding specific phrases: These are fairly common and we don't find them exceedingly informal. E.g. Steiner et al. (2022) also uses "We then proceed..." and "We can see..."

---

> > > > ### Comment · Reviewer_PVtk · 2026-04-14
> > > >
> > > > Thank you for the detailed response.
> > > >
> > > > After reading your rebuttal, the central concerns raised in my review remain.
> > > >
> > > > To clarify my position, the paper centers on a relatively narrow implementation discrepancy between two deep learning libraries. My review is not arguing that the study is entirely without merit; my concern is that the work does not establish sufficient significance for publication in its current form. BetaCrop-related parameterizations have been explored previously, and the paper mainly adds an empirical study along this axis. This makes the contribution depend less on proposing a new method and more on the strength and scope of the empirical evidence.
> > > >
> > > > Because the paper makes claims about a widely used augmentation strategy, the evidentiary burden is correspondingly higher. My core concern is that the current scope of experiments does not justify the claims and framing in the paper. The study does not convincingly demonstrate significant differences between the studied crop variants, and is not general enough to apply to commonly used training regimes and backbones.
> > > >
> > > > Likewise, my concerns about comparative baselines remain. Even if some of the methods I mentioned are not perfectly aligned with your preferred framing, the broader point is that stronger empirical context is needed to assess whether the paper materially advances the literature on crop-based augmentation.
> > > >
> > > > I also need to address the following statement from your reply:
> > > >
> > > > > We are willing to offer running 400 epoch experiments given that reviewer PVtk is interested in the regime of higher training budgets but only if reviewer PVtk commits to recommend acceptance with their inclusion.
> > > >
> > > > This is not an appropriate way to frame the review process. My comments are intended to communicate what I see as the current limitations of the paper and what kinds of additions would be needed to strengthen it. They are not a negotiation over acceptance contingent on specific experiments.

---

> > > > > ### Author Response · Authors · 2026-04-14
> > > > >
> > > > > We thank reviewer PVtk for the reply but find few concrete points raised. Therefore, our response will be limited.
> > > > >
> > > > > > The study does not convincingly demonstrate significant differences between the studied crop variants
> > > > >
> > > > > This is demonstrably false. Besides the error bars included in the results, Appendix F.2 explicitly tests for statistical significance and shows significant differences between the TensorFlow and PyTorch Inception crop.
> > > > >
> > > > > > This is not an appropriate way to frame the review process. My comments are intended to communicate what I see as the current limitations of the paper and what kinds of additions would be needed to strengthen it. They are not a negotiation over acceptance contingent on specific experiments.
> > > > >
> > > > > Editors can chime in but we don't see any guidelines against asking reviewers to make their criteria explicit. If (perhaps in combination with addressing other points) inclusion of such experiments would not change the recommendation, it is hard to argue that the reviewer really means that those additions "would be needed to strengthen it".

---

> > > > > > ### Comment · Reviewer_PVtk · 2026-04-14
> > > > > >
> > > > > > To avoid mischaracterisation:
> > > > > >
> > > > > > My point is not that statistically significant differences do not exist. To reiterate, the concern is that the paper does not establish differences that are sufficiently substantial, consistent across regimes, or general to support the level of significance suggested by the paper.
> > > > > >
> > > > > > Regarding the suggestion of additional experiments, review comments are *intended to indicate directions that could strengthen the work*. They are not a set of conditions under which acceptance would be guaranteed. The statement that "additional experiments would be provided only if there is a commitment to recommend acceptance" reflects a misunderstanding of the review process. Again, review feedback is not a negotiation of "experiments over outcomes", irrespective of whether such behaviour is explicitly codified in guidelines.

---

> > > > > > > ### Author Response · Authors · 2026-04-14
> > > > > > >
> > > > > > > We can only respond to concrete points raised.
> > > > > > >
> > > > > > > When multiple costly experiments are suggested under "Requested Changes", offering running some of them if the reviewer commits to recommend acceptance is not so much negotiation but prioritization. Perhaps [this guideline from a neighbor field](https://aclrollingreview.org/acguidelines#sources-of-information-for-meta-review) can be informative:
> > > > > > >
> > > > > > > > Using reviewer discussion to decide on revisions to request. The discussion can also serve to get the reviewers to agree on top priorities for revision when the reviews contain many suggestions. It may help to list what you see as the top opportunities for improvement and ask reviewers whether they would raise their score for a revision that satisfies those points.

---

> > > > > > > > ### Comment · Reviewer_PVtk · 2026-04-18
> > > > > > > >
> > > > > > > > The guideline you reference points to the coordination effort between reviewers and the action editor to identify priorities for revision. It does not apply to author–reviewer interaction in rebuttal, and in any case is not part of TMLR review guidelines.
> > > > > > > >
> > > > > > > > The point remains unchanged: review feedback indicates directions that may strengthen the work, but it is not a mechanism for negotiating outcomes contingent on specific additions.
> > > > > > > >
> > > > > > > > I do not have further comments on this aspect.

---

### Review · Reviewer_vjZQ · 2026-04-09

**Summary Of Contributions:**

The paper presents a detailed study of the negative impact of Tensorflow's implementation of Inception crop (IC) on the test performance of ViT-S/16 models on ImageNet-1K (IN1K) and how it can be mitigated by using BetaCrop. The study reveals important findings like how the lower limit of the distribution of the crop scale determines the augmentation strength or how training for longer requires stronger augmentation. The study also presents cases under which BetaCrop could be preferred over the Pytorch implementation of IC. Examples of such cases include situations where one does not want to tune the lower crop scale limit to maximize performance. That said, it is not entirely clear if there is a general usecase for BetaCrop unlike Pytorch IC, whose overall performance is better than BetaCrop.

**Audience:**

Yes

**Audience Explanation:**

The experimental findings of the paper and the conclusions drawn from them could be useful for researchers investigating models along similar dimensions in the future.

**Broader Impact Concerns:**

Please see Requested Changes

**Claims And Evidence:**

Yes

**Claims Explanation:**

Te claims made in the paper seem to be backed by thorough experiments across different hyperparameter settings. Moreoever, the paper also demonstrates that the claims hold true for a second optimizer.

**Requested Changes:**

It would be great if the authors could address the following concerns
* It is not very clear why one would chose to use BetaCrop when the Pytorch Implementation crop (IC) seems to do its job pretty well across different usecases
* The paper investigates a particular variant of IC implementation for a specific usecase of ImageNet classification using a specific ViT architecture, which seems pretty limited in scope, given the kind of tasks AI research is tackling these days, beyond just image classification
* It's unclear what the authors meant by "optimal settings and likely due to stronger effective augmentation" in 6.1

---

> ### Author Response · Authors · 2026-04-11
>
> We thank Reviewer vjZQ for their encouraging feedback. We answer the questions raised below.
>
> > It is not very clear why one would chose to use BetaCrop when the Pytorch Implementation crop (IC) seems to do its job pretty well across different usecases
>
> Beta crop actually performs the same or better in most settings (Table 1-3, Figure 8), especially in the case of training budget & augmentation strength mismatch. Such mismatch is not always avoidable: In some cases we may want to continue training a model for which we only allocated lower budget initially. The only case for which Beta corp underperforms is when both training budget and augmentation strengths are high. We postulate that it is due to too many crops near zero area (in agreement with reviewer tMK2) and suggest small but nonzero $a_\mathrm{min}$ for future iterations.
>
> > The paper investigates a particular variant of IC implementation for a specific usecase of ImageNet classification using a specific ViT architecture, which seems pretty limited in scope, given the kind of tasks AI research is tackling these days, beyond just image classification
>
> We acknowledge the limitation of single model architecture (adaptations for Scion optimizer notwithstanding) and supervised training setup in Sec. 8. That said, both TensorFlow and PyTorch IC are widely used for training models with different architectures and for other tasks (Sec. 1), so finding significant differences between them has implications beyond the limited scope.
>
> > It's unclear what the authors meant by "optimal settings and likely due to stronger effective augmentation" in 6.1
>
> Reviewer PVtk also raised issues with Sec. 6.1 and we agree. We have trimmed the interpretation and discussion in Sec. 6.1. We just meant that the settings in which the TensorFlow IC performs better can be explained by the differences in effective augmentation strength and now refer the readers to Appendix F.3.

---

> > ### Comment · Reviewer_vjZQ · 2026-04-20
> > **Reply to author responses**
> >
> > Thanks. The responses from the authors address my questions.

---

### Author Response · Authors · 2026-04-11
**Minor revision**

Dear AE, reviewers, and readers: based on the feedbacks we have
1. Clarified the novelty and contribution of Beta crop ("Based on 2. we revisit the performance of Beta crop" instead of proposing Beta crop)
2. Deleted the old Figure 1, which served as a teaser image but now we find it repetitive and space-inefficient
3. Trimmed the interpretation and discussion in Sec. 6.1 and clarified what we meant by "stronger effective augmentation".

---

> ### Author Response · Authors · 2026-04-21
> **Last minor revision**
>
> We double-checked the source code. The image resolution for which we take the stats of crop size distribution has been $256 \times 512$ instead of $128 \times 256$.

---

### Decision · Action_Editor_dk8f · 2026-06-28

**Recommendation:** Reject

**Audience:**

Yes

**Audience Explanation:**

Yes, researchers will be interested in data augmentation for vision model training.

**Claims And Evidence:**

No

**Claims Explanation:**

This paper highlights a hidden implementation difference between the Pytorch and TF/JAX versions of the "inception crop" which is commonly used in ImageNet classifier training. The authors find that the TF implementation oversamples smaller areas compared to pytorch, leading to minor performance degradation. The authors then propose a more flexible sampling scheme based on a beta distribution, as well as explore the effects of cropping on saliency sparsity.

While the authors conduct a reasonably thorough empirical study within their chosen setup, the reviewers reached a consensus that the evidence provided is not sufficiently convincing, substantial, or broad enough to support the paper's claims and warrant publication. Specifically, several critical shortcomings regarding the evidence were identified:
- Reviewers noted that the performance discrepancies between the TensorFlow/JAX and PyTorch implementations of the Inception crop are practically negligible. For the best-performing models trained for the longest durations, the differences are often less than 0.1 to 0.4 percentage points. Such minor improvements make it difficult to substantiate the claim that practitioners are running the "wrong" crop.
- The evidence is heavily restricted to supervised training of a small-capacity model (ViT-S) on ImageNet-1k for relatively short training budgets ( < 300 epochs). Reviewers pointed out that this does not reflect modern vision training regimes, which often utilize larger backbones and longer schedules (e.g., 400 to 800 epochs), nor does it explore downstream tasks like object detection or semantic segmentation where cropping strategies differ significantly.
- The study fails to compare the proposed BetaCrop against other established cropping and augmentation strategies in the literature (such as SRC or RICAP), making it difficult to assess whether BetaCrop provides a meaningful advancement.
- The evidence does not convincingly demonstrate a general use case for BetaCrop. It adds hyperparameter complexity without materializing a consistent or significant accuracy benefit over the standard PyTorch implementation, which already performs well across various settings.
- The claims regarding saliency sparsification were viewed as primarily observational. Because they rely on specific interpretability methods (like GradCAM and LRP) without providing a deeper mechanistic link to the augmentation findings, the evidence here was deemed orthogonal and insufficiently robust.

For the reubttal, while the authors provided detailed textual responses and clarified some misunderstandings, they largely pushed back against the reviewers' requests for additional experiments rather than providing the requested evidence.